# A Sparse Interactive Model for Matrix Completion with Side Information

**Jin Lu**     **Guannan Liang**     **Jiangwen Sun**     **Jinbo Bi**
University of Connecticut
Storrs, CT 06269
{jin.lu, guannan.liang, jiangwen.sun, jinbo.bi}@uconn.edu

## Abstract

Matrix completion methods can benefit from side information besides the partially observed matrix. The use of side features that describe the row and column entities of a matrix has been shown to reduce the sample complexity for completing the matrix. We propose a novel sparse formulation that explicitly models the interaction between the row and column side features to approximate the matrix entries. Unlike early methods, this model does not require the low rank condition on the model parameter matrix. We prove that when the side features span the latent feature space of the matrix to be recovered, the number of observed entries needed for an exact recovery is $O(\log N)$ where $N$ is the size of the matrix. If the side features are corrupted latent features of the matrix with a small perturbation, our method can achieve an $\epsilon$-recovery with $O(\log N)$ sample complexity. If side information is useless, our method maintains a $O(N^{3/2})$ sampling rate similar to classic methods. An efficient linearized Lagrangian algorithm is developed with a convergence guarantee. Empirical results show that our approach outperforms three state-of-the-art methods both in simulations and on real world datasets.

## 1   Introduction

Matrix completion has been a basis of many machine learning approaches for computer vision [6], recommender systems [21, 24], signal processing [19, 27], and among many others. Classically, low-rank matrix completion methods are based on matrix decomposition techniques which require only the partially observed data in the matrix [15, 3, 14] by solving the following problem

$$\min_{\mathbf{E}} \|\mathbf{E}\|_*, \quad \text{subject to} \quad R_\Omega(\mathbf{E}) = R_\Omega(\mathbf{F}), \tag{1}$$

where $\mathbf{F} \in \mathbb{R}^{m \times n}$ is the partially observed low-rank matrix (with a rank of $r$) that needs to be recovered, $\Omega \subseteq \{1, \cdots, m\} \times \{1, \cdots, n\}$ be the set of indexes where the corresponding components in $\mathbf{F}$ are observed, the mapping $R_\Omega(\mathbf{M})$: $\mathbb{R}^{m \times n} \to \mathbb{R}^{m \times n}$ gives another matrix whose $(i, j)$-th entry is $\mathbf{M}_{i,j}$ if $(i, j) \in \Omega$ (or 0 otherwise), and $\|\mathbf{E}\|_*$ computes the nuclear norm of $\mathbf{E}$. Early theoretical analysis [4, 5, 20] proves that $O(Nr \log^2 N)$ entries are sufficient for an exact recovery if the observed entries are uniformly sampled at random where $N = \max\{n, m\}$.

Recent studies start to explore side information for matrix completion and factorization [1, 18, 7, 17, 8]. For example, to infer the missing ratings in a user-movie rating matrix, descriptors of the users and movies are often known and may help to build a content-based recommender system. For instance, kids tend to like cartoons, so the age of a user likely interacts with the cartoon feature of a movie. When few ratings are known, this side information could be the main source for completing the matrix. Although based on empirical studies, several works found that side features are helpful [17, 18], those methods are based on non-convex matrix factorization formulations without any theoretical guarantees. Three recent methods have focussed on convex nuclear-norm regularized objectives, which leads to theoretical guarantees on matrix recovery [13, 28, 9, 16]. These methods all construct an inductive model $\mathbf{X}^T \mathbf{G} \mathbf{Y}$ so that $R_\Omega(\mathbf{X}^T \mathbf{G} \mathbf{Y}) = R_\Omega(\mathbf{F})$ where the side matrices

$\mathbf{X}$ and $\mathbf{Y}$ consist of side features, respectively, for the row entities (e.g., users) and column entities (e.g., movies) of a (rating) matrix. This inductive model has a parameter matrix $\mathbf{G}$ which is either required to be low rank [13] or to have a minimal nuclear norm $\|\mathbf{G}\|_*$ [28]. Recovering $\mathbf{G}$ of a (usually) smaller size is argued to be easier than directly recovering the matrix $\mathbf{F}$. With a very strong assumption on 'perfect' side information, i.e., both $\mathbf{X}$ and $\mathbf{Y}$ are orthonormal matrices and respectively in the latent column and row space of the matrix $\mathbf{F}$, the method in [28] is proved to require much reduced sample complexity $O(\log N)$ for an exact recovery of $\mathbf{F}$. Because most side features $\mathbf{X}$ and $\mathbf{Y}$ are not perfect in practice, a very recent work [9] proposes to use a residual matrix $\mathbf{N}$ to handle the noisy side features. This method constructs an inductive model $\mathbf{X}^T\mathbf{GY} + \mathbf{N}$ to approximate $\mathbf{F}$ and requires both $\mathbf{G}$ and $\mathbf{N}$ to be low rank, or have a low nuclear norm. It uses the nuclear norm of the residual to quantify the usefulness of side information, and proves $O(\log N)$ sampling rate for an $\epsilon$-recovery when $\mathbf{X}$ and $\mathbf{Y}$ span the full latent feature space of $\mathbf{F}$, and $o(N)$ sample complexity when $\mathbf{X}$ and $\mathbf{Y}$ contain corrupted latent features of $\mathbf{F}$. An $\epsilon$-recovery is defined as that the expected discrepancy between the predicted matrix and the true matrix is less than an arbitrarily small $\epsilon > 0$ under a certain probability.

In this paper, we propose a new method for matrix recovery by constructing a sparse interactive model $\mathbf{X}^T\mathbf{GY}$ to approximate $\mathbf{F}$ where $\mathbf{G}$ can be sparse but does not need to be low rank. The $(i, j)$-th element of $\mathbf{G}$ determines the role of the interaction between the $i$-th feature of users and the $j$-th feature of products. The low-rank property of $\mathbf{F}$ is commonly assumed to characterize the observation that similar users tend to rate similar products similarly [4]. When using an inductive approximation $\mathbf{F} = \mathbf{X}^T\mathbf{GY}$, $\text{rank}(\mathbf{F}) \leq \text{rank}(\mathbf{G})$, so a low-rank requirement on $\mathbf{G}$ can be a sufficient condition on the low-rank condition of $\mathbf{F}$. Previous relevant methods [13, 28, 9] all impose the low-rank condition on $\mathbf{G}$, which is however not a necessary condition for $\mathbf{F}$ to be low rank (only becomes a necessary condition when $\mathbf{X}$ and $\mathbf{Y}$ are full rank). Given general side matrices $\mathbf{X} \in \mathbb{R}^{d_1 \times m}$ and $\mathbf{Y} \in \mathbb{R}^{d_2 \times n}$ where the numbers of features $d_1$, $d_2 \ll N$, limiting the interactive model of $\mathbf{G} \in \mathbb{R}^{d_1 \times d_2}$ to be low rank can be an over-restrictive constraint. In our model, we use a low-rank matrix $\mathbf{E}$ to directly approximate $\mathbf{F}$ and then estimate $\mathbf{E}$ from the interactive model of $\mathbf{X}$ and $\mathbf{Y}$ with a sparse regularizer on $\mathbf{G}$. We show empirically that a low-rank $\mathbf{F}$ can be recovered from a corresponding full (or high) rank $\mathbf{G}$. Our contributions are summarized as follows:

(i) We propose a new formulation that estimates both $\mathbf{E}$ and $\mathbf{G}$ by imposing a nuclear-norm constraint on $\mathbf{E}$ but a general regularizer on $\mathbf{G}$, e.g., the sparse regularizer $\|\mathbf{G}\|_1$. The proposed model has recovery guarantees depending on the quality of the side features: (1) when $\mathbf{X}$ and $\mathbf{Y}$ are full row rank and span the entire latent feature space of $\mathbf{F}$ (but are not required to satisfy the much stronger condition of being orthonormal as in [28]), $O(\log N)$ observations are still sufficient for our method to achieve an exact recovery of $\mathbf{F}$. (2) When the side matrices are not full rank and corrupted from the original latent features of $\mathbf{F}$, i.e., $\mathbf{X}$ and $\mathbf{Y}$ do not contain enough basis to exactly recover $\mathbf{F}$, $O(\log N)$ observed entries can be sufficient for an $\epsilon$-recovery.

(ii) A new linearized alternating direction method of multipliers (LADMM) is developed to efficiently solve the proposed formulation. Existing methods that use side information are solved by standard block-wise coordinate descent algorithms which have convergence guarantee to a global solution only when each block-wise subproblem has a unique solution [26]. Our LADMM has stronger convergence property [29] and benefits from the linear convergence rate of ADMM [11, 23].

(iii) Prior methods focus on the recovery of $\mathbf{F}$, and little light has been shed to understand whether the interactive model of $\mathbf{G}$ can be retrieved. Because of the explicit use of $\mathbf{E}$ and $\mathbf{G}$, our method aims to directly recover both. The unique $\mathbf{G}$ in the case of exact recovery of $\mathbf{F}$ can be attained by our algorithm. When $\mathbf{G}$ is not unique in the $\epsilon$-recovery case, our algorithm converges to a point in the optimal solution set.

## 2 The Proposed Interactive Model

To utilize the side information in $\mathbf{X}$ and $\mathbf{Y}$ to complete $\mathbf{F}$, we consider to build a predictive model from the observed components that predicts the missing ones. One can simply build a linear model: $f = \mathbf{x}^T\mathbf{u} + \mathbf{y}^T\mathbf{v} + g$, where $\mathbf{x}$ and $\mathbf{y}$ are the feature vectors respectively for a user and a product, and $\mathbf{u}$, $\mathbf{v}$ and $g$ are model parameters. In real life applications, interactive terms between the features in $\mathbf{X}$ and $\mathbf{Y}$ can be very important. For example, male users tend to rate science fiction and action movies higher than female, which can be informative when predicting their ratings. Therefore, a linear model considering no interactive terms can be oversimple and have low predictive power for missing entries. We hence add interactive terms by introducing an interaction matrix $\mathbf{H}^{d_1 \times d_2}$ into the predictive model, which can be written as: $f = \mathbf{x}^T\mathbf{Hy} + \mathbf{x}^T\mathbf{u} + \mathbf{y}^T\mathbf{v} + g$. By defining

$\bar{\mathbf{x}} = [\mathbf{x}^T \ 1]^T$, $\bar{\mathbf{y}} = [\mathbf{y}^T \ 1]^T$ and $\mathbf{G}^{(a=d_1+1)\times(b=d_2+1)} = \begin{pmatrix} \mathbf{H} & \mathbf{u} \\ \mathbf{v}^T & g \end{pmatrix}$ the above model can be simplified to: $f = \bar{\mathbf{x}}^T \mathbf{G} \bar{\mathbf{y}}$. The following optimization problem can be solved to obtain the model parameter $\mathbf{G}$.

$$\min_{\mathbf{G},\mathbf{E}} \quad g(\mathbf{G}) + \lambda_E \|\mathbf{E}\|_*, \qquad \text{subject to} \quad \bar{\mathbf{X}}^T \mathbf{G} \bar{\mathbf{Y}} = \mathbf{E}, R_\Omega(\mathbf{E}) = R_\Omega(\mathbf{F}),$$

where $\mathbf{E}$ is a completed version of $\mathbf{F}$, $\bar{\mathbf{X}}^{a\times m}$ and $\bar{\mathbf{Y}}^{b\times n}$ are two matrices that are created by augmenting one row of all ones to $\mathbf{X}$ and to $\mathbf{Y}$, respectively, and $g(\mathbf{G})$ and $\|\mathbf{E}\|_*$ are used to incorporate the (sparsity) prior of $\mathbf{G}$ and low rank prior of $\mathbf{E}$. Because the side information data can be noisy and not all the features and their interactions are helpful to the prediction of $\mathbf{F}$, a sparse $\mathbf{G}$ is often expected. Our implementation has used $g(\mathbf{G}) = \|\mathbf{G}\|_1$. It is natural to impose low rank requirement on $\mathbf{E}$ because it is a completed version of a low rank matrix $\mathbf{F}$. The tuning parameter $\lambda_E$ is used to balance the two priors in the objective.

Without loss of generality and for convenience of notation, we simply use $\mathbf{X}$ and $\mathbf{Y}$ to denote the augmented matrices. Denote the Frobenius norm of a matrix by $\|\cdot\|_F$. To account for Gaussian noise, we relax the equality constraint $\mathbf{X}^T \mathbf{G} \mathbf{Y} = \mathbf{E}$ and replace it by minimizing their squared residual: $\|\mathbf{X}^T \mathbf{G} \mathbf{Y} - \mathbf{E}\|_F^2$ and solve the following convex optimization problem to obtain $\mathbf{G}$ and $\mathbf{E}$:

$$\min_{\mathbf{G},\mathbf{E}} \quad \frac{1}{2}\|\mathbf{X}^T \mathbf{G} \mathbf{Y} - \mathbf{E}\|_F^2 + \lambda_G g(\mathbf{G}) + \lambda_E \|\mathbf{E}\|_*, \qquad \text{subject to} \quad R_\Omega(\mathbf{E}) = R_\Omega(\mathbf{F}). \quad (2)$$

where $\lambda_G$ is another tuning parameter that together with $\lambda_E$ balances the three terms in the objective. Especially, the regularizer $g(\cdot)$ in our theoretical analysis can take any general matrix norm that satisfies $\|\mathbf{M}\|_* \leq C g(\mathbf{M})$, $\forall \mathbf{M}$, for a constant $C$, so for instance $g(\cdot)$ can be $\|\mathbf{G}\|_1$, or $\|\mathbf{G}\|_F$, or $\|\mathbf{G}\|_2$. Throughout this paper, the matrices $\mathbf{X}$ (and $\mathbf{Y}$) refer to, i.e., either the original $\mathbf{X}^{d_1\times m}$ (and $\mathbf{Y}^{d_2\times n}$) or the augmented $\bar{\mathbf{X}}^{a\times m}$ (and $\bar{\mathbf{Y}}^{b\times n}$) depending on the user-specified model.

Our formulation (2) differs from existing methods that make use of side information for matrix completion in several ways. Existing methods [28, 13, 9] solve the problem by finding $\hat{\mathbf{H}}$ that minimizes $\|\mathbf{H}\|_*$ subject to $R_\Omega(\mathbf{X}^T \mathbf{H} \mathbf{Y}) = R_\Omega(\mathbf{F})$, but we expand it to include the linear term within the interactive model. The proposed model adds the flexibility to consider both linear and quadratically interactive terms, and allows the algorithm to determine the terms that should be used in the model by enforcing the sparsity in $\mathbf{H}$ (or $\mathbf{G}$). Because $\mathbf{E} = \mathbf{X}^T \mathbf{G} \mathbf{Y}$, the rank of $\mathbf{G}$ bounds that of $\mathbf{E}$ from above. The existing methods all control the rank of $\mathbf{G}$ (e.g. by minimizing $\|\mathbf{G}\|_*$) to incorporate the prior of low rank $\mathbf{E}$ (and thus low rank $\mathbf{F}$) in their formulations. However, when the rank of $\mathbf{G}$ is not properly chosen during the tuning of hyperparameters, it may not even be a sufficient condition to ensure low rank $\mathbf{E}$ (if rank($\mathbf{E}$) $\ll$ the pre-specified rank($\mathbf{G}$)). It is easy to see that besides $\mathbf{G}$ a low-rank $\mathbf{X}$ or $\mathbf{Y}$ can lead to a low-rank $\mathbf{E}$ as well. Enforcing a low-rank condition on $\mathbf{H}$ or $\mathbf{G}$ may limit the search space of the interactive model and thus impair the prediction performance on missing matrix entries, which are demonstrated in our empirical results. Moreover, one can observe that when $\lambda_G$ is sufficiently large, Eq.(2) is reduced to the standard matrix completion problem (1) without side information because $\mathbf{G}$ may be degenerated into a zero matrix, so our formulation is applicable when no access to useful side information.

## 3 Recovery Analysis

Let $\mathbf{E}_0$ and $\mathbf{G}_0$ be the two matrices such that $R_\Omega(\mathbf{F}) = R_\Omega(\mathbf{E}_0)$ and $\mathbf{E}_0 = \mathbf{X}^T \mathbf{G}_0 \mathbf{Y}$. In this section, we give our theoretical results on the sample complexity for achieving an exact recovery of $\mathbf{E}_0$ and $\mathbf{G}_0$ when $\mathbf{X}$ and $\mathbf{Y}$ are both full row rank (i.e., rank($\mathbf{X}$) $= a$ and rank($\mathbf{Y}$) $= b$), and an $\epsilon$-recovery of $\mathbf{E}_0$ when the two side matrices are corrupted and less informative. The proofs of all theorems are given in supplementary materials.

### 3.1 Sample Complexity for Exact Recovery

Before presenting our results, we give a few definitions. Let $\mathbf{F} = \mathbf{U}\boldsymbol{\Sigma}\mathbf{V}^T$, $\mathbf{X}^T = \mathbf{U_X}\boldsymbol{\Sigma_X}\mathbf{V_X}^T$ and $\mathbf{Y}^T = \mathbf{U_Y}\boldsymbol{\Sigma_Y}\mathbf{V_Y}^T$ be the singular value decomposition of $\mathbf{F}$, $\mathbf{X}^T$ and $\mathbf{Y}^T$, respectively, where all $\boldsymbol{\Sigma}$ matrices are full rank, meaning that singular vectors corresponding to the singular value 0 are not included in the respective $\mathbf{U}$ and $\mathbf{V}$ matrices. Let

$$P_{\mathbf{U}} = \mathbf{U}\mathbf{U}^T \in \mathbb{R}^{m\times m}, \quad P_{\mathbf{V}} = \mathbf{V}\mathbf{V}^T \in \mathbb{R}^{n\times n},$$

$$P_{\mathbf{X}} = \mathbf{U_X}\mathbf{U_X}^T = \mathbf{X}^T\mathbf{V_X}\boldsymbol{\Sigma_X}^{-2}\mathbf{V_X}^T\mathbf{X} \in \mathbb{R}^{m\times m}, \quad P_{\mathbf{Y}} = \mathbf{U_Y}\mathbf{U_Y}^T = \mathbf{Y}^T\mathbf{V_Y}\boldsymbol{\Sigma_Y}^{-2}\mathbf{V_Y}^T\mathbf{Y} \in \mathbb{R}^{n\times n},$$

where $P_{\mathbf{U}}$, $P_{\mathbf{V}}$, $P_{\mathbf{X}}$ and $P_{\mathbf{Y}}$ project a vector onto the subspaces spanned, respectively, by the columns in $\mathbf{U}$, $\mathbf{V}$ and rows in $\mathbf{X}$, and $\mathbf{Y}$. For any matrix $\mathbf{M}^{m \times n}$ that satisfies $\mathbf{M} = P_{\mathbf{X}}\mathbf{M}P_{\mathbf{Y}}$, we define two linear operators: $P_T : \mathbb{R}^{m \times n} \to \mathbb{R}^{m \times n}$ and $P_{T^\perp} : \mathbb{R}^{m \times n} \to \mathbb{R}^{m \times n}$ as follows:

$$P_T(\mathbf{M}) = \; P_{\mathbf{U}}\mathbf{M}P_{\mathbf{Y}} + P_{\mathbf{X}}\mathbf{M}P_{\mathbf{V}} - P_{\mathbf{U}}\mathbf{M}P_{\mathbf{V}}$$
$$P_{T^\perp}(\mathbf{M}) = \; (P_{\mathbf{X}} - P_{\mathbf{U}})\mathbf{M}(P_{\mathbf{Y}} - P_{\mathbf{V}}) = P_{\mathbf{X}^\perp}\mathbf{M}P_{\mathbf{Y}^\perp}.$$

Let $\mu_0$ and $\mu_1$ be the two coherence measures of $\mathbf{F}$ and be defined as follows as discussed in [4, 16]:

$$\mu_0 = \max\left(\frac{m}{r} \max_{1 \leq i \leq m} \|P_{\mathbf{U}}\mathbf{e}_i\|_2, \frac{n}{r} \max_{1 \leq j \leq n} \|P_{\mathbf{V}}\mathbf{e}_j\|_2\right), \quad \mu_1 = \max_{i,j} \frac{mn}{r}([\mathbf{U}\mathbf{V}^T]_{i,j})^2,$$

where $\mathbf{e}_i$ is the unit vector with the $i$th entry equal to 1. Let $\mu_{\mathbf{XY}}$ be the coherence measurement between $\mathbf{X}$ and $\mathbf{Y}$ and be defined as:

$$\mu_{\mathbf{XY}} = \max\left(\max_{1 \leq i \leq m} \frac{m\|\mathbf{x}_i\|_2^2}{a}, \max_{1 \leq j \leq n} \frac{n\|\mathbf{y}_j\|_2^2}{b}\right).$$

With the above definitions, we show in the following theorem that when $\mathbf{X}$ and $\mathbf{Y}$ are both full row rank, $(\mathbf{G}_0, \mathbf{E}_0)$ is the unique solution to Eq.(2) with high probability as long as there are $O(r \log N)$ observed components in $\mathbf{F}$. In other words, with a sampling rate of $O(r \log N)$, our method can fully recover both $\mathbf{E}_0$ and $\mathbf{G}_0$ with a high probability when $\mathbf{X}$ and $\mathbf{Y}$ are full row rank.

**Theorem 1** *Let $\mu = \max(\mu_0, \mu_{\mathbf{XY}})$, $\sigma = \max(\|\mathbf{\Sigma}_{\mathbf{X}}^{-1}\|_*, \|\mathbf{\Sigma}_{\mathbf{Y}}^{-1}\|_*)$, $N = \max(m, n)$, $q_0 = \frac{1}{2}(1 + \log a - \log r)$, $T_0 = \frac{128p}{3}\sigma\mu \max(\mu_1, \mu)r(a + b) \log N$ and $T_1 = \frac{8p}{3}\sigma^2\mu^2(ab + r^2) \log N$, where $p$ is a constant. Assume $T_1 \geq q_0 T_0$, $\mathbf{X}$ and $\mathbf{Y}$ are both full row rank. For any $p > 1$, with a probability at least $1 - 4(q_0 + 1)N^{-p+1} - 2q_0 N^{-p+2}$, $(\mathbf{G}_0, \mathbf{E}_0)$ is the unique optimizer to Problem (2) with necessary sampling rate as few as $O(r \log N)$. More precisely, the sampling size $|\Omega|$ should satisfy that $|\Omega| \geq \frac{64p}{3}\sigma\mu \max(\mu_1, \mu)(1 + \log a - \log r)r(a + b) \log N$.*

When $r \ll N$ and $r = O(1)$, the sampling rate for the exact recovery of both $\mathbf{E}_0$ and $\mathbf{G}_0$ reduces to $O(\log N)$. A similar sampling rate for a full recovery of $\mathbf{E}_0$ has been developed in [28] where both $\mathbf{X}$ and $\mathbf{Y}$, however, need to be orthonormal matrices in their derivation. In Theorem 1, because $\sigma$ is mainly determined by the smallest singular values of the side information matrices, and sampling rate increases when $\sigma$ increases, it suggests that side information matrices of lower rank would require more observed $\mathbf{F}$ entries for a full recovery of $\mathbf{F}$. An advanced model without the orthonormal assumption has been given in [9], but exact recovery is not discussed. In our case, the two matrices are only required to be full row rank. Moreover, the theoretical or empirical results in our work give the first careful investigation on the recovery of both $\mathbf{G}_0$ and $\mathbf{E}_0$.

### 3.2 Sample Complexity for $\epsilon$-Recovery

The condition for full-rank side information matrices may not be satisfied in some cases to fully recover $\mathbf{E}_0$ (or $\mathbf{F}$). We analyze the error bound of our model and prove a reduced sample complexity in comparison with standard matrix completion methods for an $\epsilon$-recovery when the side information matrices are not full row rank or their rank is difficult to attain.

**Theorem 2** *Denote $\|\mathbf{E}\|_* \leq \alpha$, $\|\mathbf{G}\|_1 \leq \gamma$, $\|\mathbf{X}^T\mathbf{G}\mathbf{Y} - \mathbf{E}\|_F \leq \phi$ and the perfect side feature matrices (containing latent features of $\mathbf{F}$) are corrupted with $\Delta\mathbf{X}$ and $\Delta\mathbf{Y}$ where $\|\Delta\mathbf{X}\|_F \leq s_1, \|\Delta\mathbf{Y}\|_F \leq s_2$ and $S = \max(s_1, s_2)$. To $\epsilon$-recover $\mathbf{F}$ that the expected loss $\mathbb{E}[l(f, \mathbf{F})] < \epsilon$ for a given arbitrarily small $\epsilon > 0$, $O(\min((\gamma^2 + \phi^2) \log N, S^2\alpha\sqrt{N})/\epsilon^2)$ observations are sufficient for our model when corrupted factors of side information are bounded.*

Theorem 2 can be inferred from the fact that the trace norm of $\mathbf{E}$ and the $\ell_1$-norm of $\mathbf{G}$ affect sample complexity of our model. It meets the intuition that higher rank matrix ought to require more observations to recover. Besides, for the discovery of $\mathbf{G}$, a sparse interactive matrix can lead to the decrease of the sample complexity, which implies that the side information, even though when it is not perfect, could be informative enough such that the original matrix can be compressed by sparse coding via the estimated interaction between the features of row and column entities of the matrix. Our empirical evaluations have confirmed the utility of even imperfect side features.

When the rank of the original data matrix $r = O(1)$ ($r \ll N$), and correspondingly $\alpha = O(1)$, Theorem 2 points out that only $O(\log N)$ sampling rate is required for an $\epsilon$-recovery. The classic matrix completion analysis without side information shows that under certain conditions, one

can achieve $O(Npoly\log N)$ sample complexity for both perfect recovery [4] and $\epsilon$-recovery [25], which is higher than our complexity. However, the condition for these existing bounds is that the observed entries follow a certain distribution. Recent studies [22] found that if no specific distribution is pre-assumed for observed entries, $O(N^{3/2})$ sampling rate is sufficient for an $\epsilon$-recovery. Compared to those results, our analysis does not require any assumption on the distribution of observed entries. When $\mathbf{X}$ and $\mathbf{Y}$ contain insufficient interaction information about $\mathbf{F}$ and $\|\mathbf{E}\|_* = O(N)$, the sample complexity of our method increases to $O(N^{3/2})$ in the worst case, which means that our model maintains the same complexity as the classic methods.

## 4 Adaptive LADMM Algorithm

In this section, we develop an adaptive LADMM algorithm [29] to solve problem (2). First, we show that the ADMM is applicable in our problem and we then derive LADMM steps. A convergence proof is established to guarantee the performance of our algorithm.

Because it requires separable blocks of variables in order to use ADMM, we first define $\mathbf{C} = \mathbf{E} - \mathbf{X}^T\mathbf{G}\mathbf{Y}$ and use it in Eq.(2). Then the augmented Lagrangian function of (2) is given by

$$\mathcal{L}(\mathbf{E}, \mathbf{G}, \mathbf{C}, \mathbf{M}_1, \mathbf{M}_2, \beta) = \frac{1}{2}\|\mathbf{C}\|_F^2 + \lambda_E\|\mathbf{E}\|_* + \lambda_G\|\mathbf{G}\|_1 + \langle\mathbf{M}_1, R_\Omega(\mathbf{E} - \mathbf{F})\rangle +$$
$$+ \langle\mathbf{M}_2, \mathbf{E} - \mathbf{X}^T\mathbf{G}\mathbf{Y} - \mathbf{C}\rangle + \frac{\beta}{2}\|R_\Omega(\mathbf{E} - \mathbf{F})\|_F^2 + \frac{\beta}{2}\|\mathbf{E} - \mathbf{X}^T\mathbf{G}\mathbf{Y} - \mathbf{C}\|_F^2 \quad (3)$$

where $\mathbf{M}_1, \mathbf{M}_2 \in \mathbb{R}^{m\times n}$ are Lagrange multipliers and $\beta > 0$ is the penalty parameter. Given $\mathbf{C}^k$, $\mathbf{G}^k, \mathbf{E}^k, \mathbf{M}_1^k$ and $\mathbf{M}_2^k$ at iteration $k$, each group of the variables yields their respective subproblems:

$$\mathbf{C}^{k+1} = \arg\min_{\mathbf{C}} \mathcal{L}(\mathbf{E}^k, \mathbf{G}^k, \mathbf{M}_2^k, \mathbf{C}, \beta_k),$$
$$\mathbf{G}^{k+1} = \arg\min_{\mathbf{G}} \mathcal{L}(\mathbf{E}^k, \mathbf{G}, \mathbf{M}_2^k, \mathbf{C}^{k+1}, \beta_k), \quad (4)$$
$$\mathbf{E}^{k+1} = \arg\min_{\mathbf{E}} \mathcal{L}(\mathbf{E}, \mathbf{G}^{k+1}, \mathbf{M}_1^k, \mathbf{M}_2^k, \mathbf{C}^{k+1}, \beta_k),$$

After solving these subproblems, we update the multipliers $\mathbf{M}_1$ and $\mathbf{M}_2$ as follows;

$$\mathbf{M}_1^{k+1} = \mathbf{M}_1^k + \beta_k(R_\Omega(\mathbf{E}^{k+1} - \mathbf{F})),$$
$$\mathbf{M}_2^{k+1} = \mathbf{M}_2^k + \beta_k(\mathbf{E}^{k+1} - \mathbf{X}^T\mathbf{G}^{k+1}\mathbf{Y} - \mathbf{C}^{k+1}). \quad (5)$$

We focus on demonstrating the iterative steps of the adaptive LADMM. Given $\mathbf{C}^k$, $\mathbf{G}^k$ $\mathbf{E}^k$, $\mathbf{M}_1^k$ and $\mathbf{M}_2^k$, Algorithm 1 describes how to obtain the next iterate $(\mathbf{C}, \mathbf{E}, \mathbf{G}, \mathbf{M}_1, \mathbf{M}_2)$. A closed-form solution has been derived for each subproblem in the supplementary material.

---

**Algorithm 1** The adaptive LADMM algorithm to solve $\mathbf{C}^k$, $\mathbf{G}^k$, $\mathbf{E}^k$, $k = 1, ..., K$

---

**Input:** $\mathbf{X}, \mathbf{Y}$ and $R_\Omega(\mathbf{F})$ with parameters $\lambda_G$, $\lambda_E$, $\tau_A$, $\tau_B$, $\rho$ and $\beta_{max}$.
**Output:** $\mathbf{C}, \mathbf{G}, \mathbf{E}$;
1: Initialize $\mathbf{E}^0, \mathbf{G}^0, \mathbf{M}_1^0, \mathbf{M}_2^0$. Compute $\mathbf{A} = \mathbf{Y}^T \otimes \mathbf{X}^T$. $k = 0$,
   repeat;
2: $\mathbf{C}^{k+1} = \frac{\beta_k}{\beta_k+1}(\mathbf{E}^k - \mathbf{X}^T\mathbf{G}^k\mathbf{Y} + \mathbf{M}_2^k/\beta_k)$;
3: $\mathbf{G}^{k+1} = reshape(\max(|\mathbf{g}^k - f_1^k/\tau_A| - \frac{\lambda_G}{\tau_A\beta_k}, 0) \odot sgn(\mathbf{g}^k - f_1^k/\tau_A))$ where $f_1^k = \mathbf{A}^T(\mathbf{A}\mathbf{g}^k + \mathbf{c}^k - \mathbf{b}_1^k) = \mathbf{A}^T(\mathbf{A}\mathbf{g}^k + \mathbf{c}^k - \mathbf{e}^k - \mathbf{m}_2^k/\beta_k)$ and $\mathbf{e} = vec(\mathbf{E})$, $\mathbf{g} = vec(\mathbf{G})$, $\mathbf{m} = vec(\mathbf{M})$, $\mathbf{c} = vec(\mathbf{C})$.
4: $\mathbf{E}^{k+1} = SVT(\mathbf{E}^k - (f_2^k + f_3^k)/(2\tau_B), \lambda_E/2(\beta_k\tau_B))$ where $f_2^k = R_\Omega(\mathbf{E}^k - \mathbf{F} + \mathbf{M}_1^k/\beta_k)$; $f_3^k = \mathbf{E}^k - \mathbf{X}^T\mathbf{G}^{k+1}\mathbf{Y} - \mathbf{C}^k + \mathbf{M}_2^k/\beta_k$.
5: $\mathbf{M}_1^{k+1} = \mathbf{M}_1^k + \beta_k(R_\Omega(\mathbf{E}^{k+1} - \mathbf{F}))$.
6: $\mathbf{M}_2^{k+1} = \mathbf{M}_2^k + \beta_k(\mathbf{E}^{k+1} - \mathbf{X}^T\mathbf{G}^{k+1}\mathbf{Y} - \mathbf{C}^{k+1})$.
7: $\beta_{k+1} = \min(\beta_{\max}, \rho\beta_k)$.
8: $k = k + 1$ until convergence;
   Return $\mathbf{C}, \mathbf{G}, \mathbf{E}$;

---

The adaptive parameter in Algorithm 1 is $\rho > 1$, and $\beta_{\max}$ controls the upper bound of $\{\beta_k\}$. The operator $reshape(\mathbf{g})$ converts a vector $\mathbf{g} \in \mathbb{R}^{ab}$ into a matrix $\mathbf{G} \in \mathbb{R}^{a\times b}$, which is the inverse

operator of vec($\mathbf{G}$). The operator $SVT(\mathbf{E}, t)$ is the singular value thresholding process defined in [3] for soft-thresholding the singular values of an arbitrary matrix $\mathbf{E}$ by a threshold $t$. The matrix $\mathbf{A} = \mathbf{Y}^T \otimes \mathbf{X}^T$ where $\otimes$ indicates the Kronecker product. In the initialization step, $\mathbf{M}_1^0, \mathbf{M}_2^0$ are randomly drawn from the standard Gaussian distribution; we initialize $\mathbf{E}_0$ and $\mathbf{G}_0$ by the iterative soft-thresholding algorithm [2] and $SVT$ operator respectively.

The adaptive LADMM can effectively solve the proposed optimization problem in several aspects. First, the convergence of the commonly-used block-wise coordinate descent (BCD) method, sometimes referred to as alternating minimization methods, requires typically that the optimization problem be strictly convex (or quasiconvex but hemivariate). The strongest result for BCD so far is established in [26] which requires the alternating subproblems to be optimized in each iteration to its *unique* optimal solution. This requirement is often restrictive in practice. Our convex (but not strictly convex) problem can be solved by the adaptive LADMM with the global convergence guarantee which is characterized in Theorem 3. Second, two of the subproblems are non-smooth due to the $\ell_1$-norm or the nuclear norm, so it can be difficult to obtain a closed-form formula to efficiently compute a solution by standard optimization tools; however, adaptive LADMM utilizes the linearization technique which leads to a closed-form solution for each linearized subproblem, and significantly enhances the efficiency of the iterative process. Third, adaptive LADMM can be practically parallelizable by a similar scheme to that of ADMM. It is also noted that the convergence rate of LADMM [11] and parallel LADMM is $O(1/k)$ [23] whereas the BCD method still lacks of clear theoretical results of its convergence rate.

**Theorem 3** *Define the operators $\mathcal{A}$ and $\mathcal{B}$ as $\mathcal{A}(\mathbf{G}) = \begin{pmatrix} 0 \\ -\mathbf{X}^T\mathbf{G}\mathbf{Y} \end{pmatrix}$, $\mathcal{B}(\mathbf{E}) = \begin{pmatrix} R_\Omega(\mathbf{E}) \\ \mathbf{E} \end{pmatrix}$, and let $\mathbf{M} = \begin{pmatrix} \mathbf{M}_1 \\ \mathbf{M}_2 \end{pmatrix}$. If $\beta_k$ is non-decreasing and upper-bounded, $\tau_A > \|\mathcal{A}\|^2$, and $\tau_B > \|\mathcal{B}\|^2$, then the sequence $\{(\mathbf{C}^k, \mathbf{G}^k, \mathbf{E}^k, \mathbf{M}^k)\}$ generated by the adaptive LADMM Algorithm 1 converges to a global minimizer of Eq. (2).*

## 5 Experimental Results

We validated our method in both simulations and the analysis of two real world datasets: MovieLens (movie rating) and NCI-DREAM (drug discovery) datasets. Three most recent matrix completion methods that also utilized side information, MAXIDE[28], IMC[13] and DirtyIMC[9], were compared against our method. The design of our experiments focused on demonstrating the effectiveness of our method in practice. The performance of all methods was measured by the relative mean squared error (RMSE) calculated on missing entries: $\|R_{\bar{\Omega}}(\mathbf{X}^T\mathbf{G}\mathbf{Y} - \mathbf{F})\|_2^2/\|R_{\bar{\Omega}}(\mathbf{F})\|_2^2$. For both synthetic and real-world datasets, we randomly set $q$ percent of the components in each observed matrix $\mathbf{F}$ to be missing. The hyperparameters $\lambda$'s and the rank of $\mathbf{G}$ (required by IMC and DirtyIMC) were tuned via the same cross validation process: *we randomly picked 10% of the given entries to form a validation set. Then models were obtained by applying each method to the remaining entries with a specific choice of $\lambda$ from $10^{-3}, 10^{-2}, ..., 10^4$. The average validation RMSE was examined by repeating the above procedure six times. The hyperparameter values that gave the best average validation RMSE were chosen for each method.* For IMC and DirtyIMC, the best rank of $\mathbf{G}$ was chosen from $= 1$ to 15 within each data split. For each choice of $q$, we repeated the above entire procedure six times and reported the average RMSE on the missing entries.

### 5.1 Synthetic Datasets

We created two different simulation tests with and without full row rank $\mathbf{X}$ and $\mathbf{Y}$. For all the synthetic datasets, we first randomly created $\mathbf{X}$ and $\mathbf{Y}$. In order to make our simulations reminiscent real situations where distributions of side features can be heterogeneous, data for each feature in both $\mathbf{X}$ and $\mathbf{Y}$ were generated according to a distribution that was randomly selected from Gaussian, Poisson and Gamma distributions. We created the sparse $\mathbf{G}$ matrices as follows. The location of the non-zero entries of $\mathbf{G}$ were randomly picked but their values were generated by multiplying a value drawn from $\mathcal{N}(0, 100)$, which we repeated several times to chose the matrices that showed full or high rank. We then generated $\mathbf{F}$ with $\mathbf{F} = \mathbf{X}^T\mathbf{G}\mathbf{Y} + \mathbf{N}$ where $\mathbf{N}$ represents noise and each component $\mathbf{N}_{i,j}$ was drawn from $\mathcal{N}(0, 1)$. For each simulated $\mathbf{F}$, we ran all methods with $q \in [10\% - 80\%]$ with an increase step of $10\%$.

We compared the different methods in three settings, which were labeled as synthetic experiment I, II and III in our results. In the first setting, the dimension of $\mathbf{X}$ and $\mathbf{Y}$ was set to $15 \times 50$ and

$20 \times 140$ and all features in these two matrices were randomly generated to make them full row rank. Both the last two settings corresponded to the second test where $\mathbf{X}$ and $\mathbf{Y}$ were not full row rank. The dimension of $\mathbf{X}$ and $\mathbf{Y}$ was set to $16 \times 50$, $21 \times 140$ and $20 \times 50$, $25 \times 140$, respectively, for these two settings where the first 15 features in $\mathbf{X}$ and 20 features in $\mathbf{Y}$ were randomly created, but the remaining features were generated by arbitrarily linear combinations of the randomly created features. For all three settings, we used 10 synthetic datasets and reported mean and standard deviation of RMSE on missing values as shown in Figure 1.

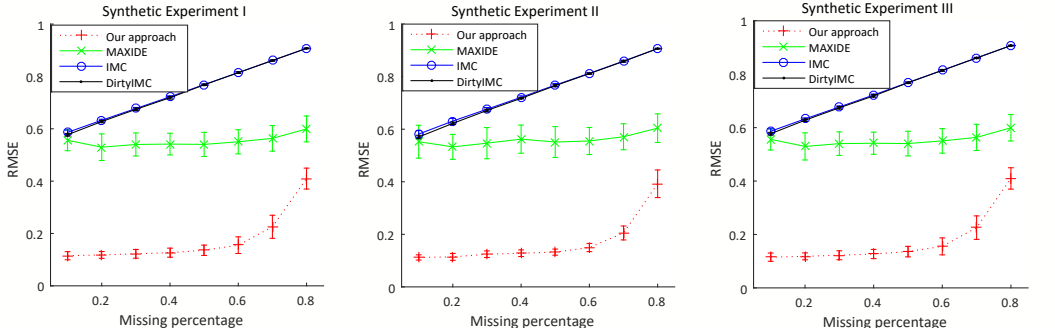

Figure 1: The Comparison of RMSE for Experiments I, II, and III.

Our approach outperformed all other compared methods significantly in almost all these settings. When the missing rate $q$ increased, the RMSE of our method grew much slower than other methods. We studied the rank of the recovered $\mathbf{G}$ and $\mathbf{E}$ in the first setting. For all methods, the corresponding $\mathbf{G}$ and $\mathbf{E}$ that gave the best performance were examined. The ranks of $\mathbf{G}$ and $\mathbf{E}$ from our method, MAXIDE, IMC, DirtyIMC were 15, 8, 1, 1 and 15, 15, 1, 2, respectively. These results suggested that incorporating the strong prior of low rank $\mathbf{G}$ might hurt the recovery performance. The retrieved model matrices $\mathbf{G}$ of all compared methods (when using $q =10\%$ of missing entries in one of the 10 synthetic datasets) together with the true $\mathbf{G}$ are plotted in Figure 2. Only our method was able to recover the true $\mathbf{G}$ and all the other methods merely found approximations.

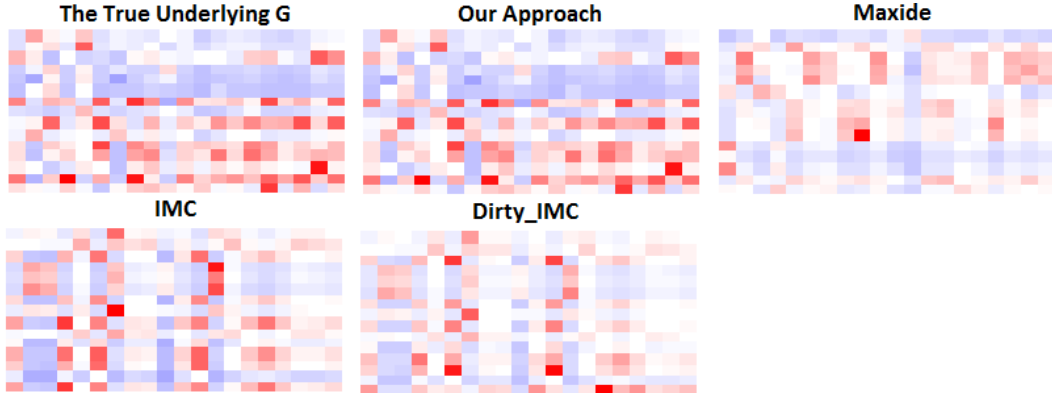

Figure 2: The heatmap of the true G and recovered G matrices in Synthetic Experiment I.

## 5.2 Real-world Datasets

We used two relatively large datasets that we could find as suitable for our empirical evaluation. Note that early methods employing side information were often tested on datasets with either $\mathbf{X}$ or $\mathbf{Y}$ but not both although some of them might be larger than the two datasets we used.

**5.2.1. MovieLens.** This dataset was downloaded from [12] and contained 100,000 user ratings (integers from 1 to 5) from 943 users on 1682 movies. There were 20 movie features such as genre and release date, as well as 24 user features describing users' demographic information such as age and gender. We compared all methods with four different $q$ values: 20-50%. The RMSE values of each method are shown in Table 1, which shows that our approach significantly outperformed other methods, especially when $q$ was large. Figure 3 shows the constructed $\mathbf{G}$ matrix that shows some interesting observations. For instance, male users tend to rate action, science fiction, thriller and war movies high but low for children' movies, exhibiting some common intuitions.

**5.2.2 NCI-DREAM Challenge.** The data on the reactions of 46 breast cancer cell lines to 26 drugs and the expression data of 18633 genes for all the cell lines were provided by NCI-DREAM Chal-

| | MovieLens Data | | | | NCI-Dream Challenge | | | |
|---|---|---|---|---|---|---|---|---|
| Methods | 20% | 30% | 40% | 50% | 20% | 30% | 40% | 50% |
| **Our approach** | **0.276** ($\pm$ **0.001**) | **0.279** ($\pm$ **0.002**) | **0.284** ($\pm$ **0.001**) | **0.292** ($\pm$ **0.001**) | **0.181** ($\pm$ **0.069**) | **0.139** ($\pm$ **0.010**) | **0.145** ($\pm$ **0.018**) | **0.190** ($\pm$ **0.031**) |
| MAXIDE | 0.424 ($\pm$0.016) | 0.425 ($\pm$0.013) | 0.419 ($\pm$0.008) | 0.421 ($\pm$0.013) | 0.268 ($\pm$0.036) | 0.240 ($\pm$0.007) | 0.255 ($\pm$0.016) | 0.288 ($\pm$0.022) |
| IMC | 0.935 ($\pm$0.001) | 0.943 ($\pm$0.001) | 0.945 ($\pm$0.001) | 0.959 ($\pm$0.001) | 0.437 ($\pm$0.031) | 0.489 ($\pm$0.003) | 0.557 ($\pm$0.013) | 0.637 ($\pm$0.011) |
| DirtyIMC | 0.705 ($\pm$0.001) | 0.738 ($\pm$0.001) | 0.775 ($\pm$0.001) | 0.814 ($\pm$0.001) | 0.432 ($\pm$0.033) | 0.475 ($\pm$0.008) | 0.551 ($\pm$0.018) | 0.632 ($\pm$0.011) |

Table 1: The Comparison of RMSE values of different methods on real-world datasets.

lenge [10]. For each drug, we had 14 features that describes their chemical and physical properties such as molecular weight, XLogP3 and hydrogen bond donor count, and were downloaded from National Center for Biotechnology Information (http://pubchem.ncbi.nlm.nih.gov/). For the cell line features, we ran principle component analysis (PCA) and used the top 45 principal components that accounted for more than 99.99% of the total data variance. We compared the four different methods with four different $q$ values: 20-50%. The RMSE values of all methods are provided in Table 1 where our method again shows the best performance. We examined the ranks of both **G** and **E** obtained by all the methods. They were 15, 15, 1, 1 for **G** and 2, 15, 1, 2 for **E**, respectively, for our approach, MAXIDE, IMC and DirtyIMC in sequence. This demonstrates that a low rank **E** but a high rank **G** give the best performance on this dataset. In other words, requiring a low rank **G** may hurt the performance of recovering a low rank **E**.

The constructed **G** by our method is plotted in Figure 4, where columns represent cell line features (i.e., principle components) and rows represent drug features. Please refer to the supplementary material for the names of these features. According to this figure, drug features: XlogP (F2), hydrogen bond donor (HBD) (F3), Hydrogen bond acceptor (HBA) (F4) and Rotatable Bond number (F5) all played important roles in drug sensitivity. This result aligns well with biological knowledge, as all these four features are very important descriptors for cellular entry and retention.

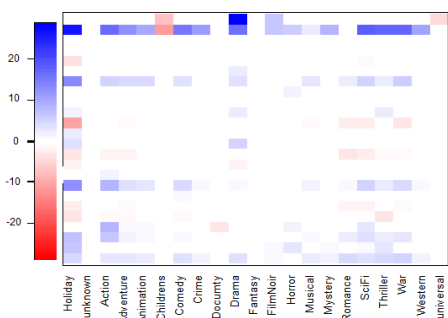

Figure 3: HeatMap of **G** for MovieLens

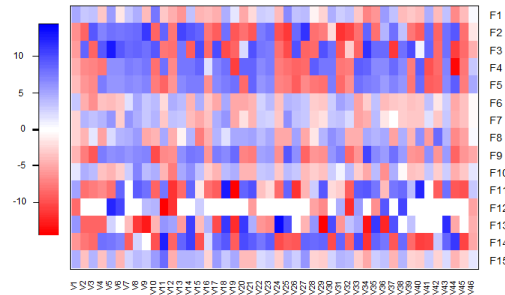

Figure 4: HeatMap of $sign(G)\log(|G|)$ for NCI-DREAM for a better illustration

## 6 Conclusion

In this paper, we have proposed a novel sparse inductive model that utilizes side features describing the row and column entities of a partially observed matrix to predict its missing entries. This method models the linear predictive power of side features as well as interaction between the features of row and column entities. Theoretical analysis shows that this model has advantages of reduced sample complexity over classical matrix completion methods, requiring only $O(\log N)$ observed entries to achieve a perfect recovery of the original matrix when the side features reflect the true latent feature space of the matrix. When the side features are less informative, our model requires $O(\log N)$ observations for an $\epsilon$-recovery of the matrix. Unlike early methods that use a BCD algorithm, we have developed a LADMM algorithm to optimize the proposed formulation. Given the optimization problem is convex, this algorithm can converge to a global solution. Computational results demonstrate the superior performance of this method over three recent methods. Future work includes the examination of other types and quality of side information and the understanding of whether our method will benefit a variety of relevant problems, such as multi-label learning, and semi-supervised clustering etc.

## Acknowledgments

Jinbo Bi and her students Jin Lu, Guannan Liang and Jiangwen Sun were supported by NSF grants IIS-1320586, DBI-1356655, and CCF-1514357 and NIH R01DA037349.

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
