[Supplementary Material · NIPS2016_cam_suppv2.pdf]

# Supplementary Materials for "A Sparse Interactive Model for Matrix Completion with Side Information"

Here we present the proof of the three theoretical parts related to our work, including the exact recovery analysis, $\epsilon$-recovery analysis and the convergence analysis. Algorithm details are followed from the theoretical part. The feature description table of the drug discovery dataset is displayed in the last section.

## 1 Exact Recovery Sampling Complexity

**Theorem 1** *Let* $\mu = \max(\mu_0, \mu_{\mathbf{XY}})$, $\sigma = \max(\|\mathbf{\Sigma}_{\mathbf{X}}^{-1}\|_*, \|\mathbf{\Sigma}_{\mathbf{Y}}^{-1}\|_*)$. $\mu_0$, $\mu_{\mathbf{XY}}$ *are calculated from* $\mathbf{F}$. *Denote* $q_0 = \frac{1}{2}(1 + \log a - \log r), T_0 = \frac{128p}{3}\sigma\mu\max(\mu_1,\mu)r(a+b)\log N$ *and* $T_1 = \frac{8p}{3}\sigma^2\mu^2(ab+r^2)\log N$. *Assume* $T_1 \geq q_0 T_0$, $\mathbf{X}$ *and* $\mathbf{Y}$ *are orthonormal. For any* $p > 1$, *with a probability at least* $1 - 4(q_0+1)N^{-p+1} - 2q_0N^{-p+2}$, $\mathbf{G}_0$ *and* $\mathbf{E}_0$ *are the unique optimizer to the problem (3) in our formulation if*

$$|\Omega| \geq \frac{64p}{3}\sigma\mu\max(\mu_1,\mu)(1+\log a - \log r)r(a+b)\log N$$

For proving Theorem 1, we introduce the Lemma 1 stating two deterministic conditions for $\mathbf{G}_0$ and $\mathbf{E}_0$ to be the unique minimizer of our problem. It can be proved in Lemma 2, Lemma 9 and Lemma 10 that under a high probability the assumption A1 and A2 hold. So let us first give Lemma 1 as below:

**Lemma 1** *We assume that for any* $\mathbf{M} \neq 0$, $\mathbf{M} \in \mathbb{R}^{m \times n}$ *satisfying* $R_{\Omega}(\mathbf{M}) = 0$ *and* $\mathbf{M} = P_{\mathbf{X}}\mathbf{M}P_{\mathbf{Y}}$, *then we have*

$$\textbf{\textit{A1}} \quad \|P_T(\mathbf{M})\|_F \leq \zeta\|P_{T^\perp}(\mathbf{M})\|_F,$$

*where*

$$\zeta \leq \sqrt{\frac{a}{2r}}$$

*And assume that there exists a matrix* $\mathbf{H} \in \mathbb{R}^{m \times n}$ *such that*

$$\textbf{\textit{A2}} \quad R_{\Omega}(\mathbf{H}) = \mathbf{H}, \|P_T(\mathbf{H}) - \mathbf{U}\mathbf{V}^T\|_F \leq \sqrt{\frac{r}{2a}}, \|P_{T^\perp}(\mathbf{H})\| < \frac{1}{2}$$

*Moreover we assume that there exists a constant* $C_0$ *such that*

$$\textbf{\textit{A3}} \quad \|\mathbf{G}_0\|_1 = s < \frac{\lambda_G(\frac{1}{2} - \zeta\sqrt{\frac{r}{2a}})}{C\lambda_E},$$

*where* $C < C_0$, *then* $\mathbf{G}_0$ *and* $\mathbf{E}_0$ *are the unique minimizer to our optimization problem.*

**Proof.** Assuming the solution is not unique, there exists another solution $\mathbf{G}_0 + \Delta\mathbf{G}$ and $\mathbf{E}_0 + \Delta\mathbf{E}$ with $\Delta\mathbf{G}, \Delta\mathbf{E} \neq 0$. Basically our aim is to prove the contradiction that $\|\mathbf{G}_0 + \Delta\mathbf{G}\|_1 + \|\mathbf{E}_0 + \Delta\mathbf{E}\|_* \geq \|\mathbf{G}_0\|_1 + \|\mathbf{E}_0\|_*$.

In order to prove the contradiction, we illustrate several useful facts below:

(1) $R_\Omega(\mathbf{X}^T(\mathbf{G}_0 + \Delta\mathbf{G})\mathbf{Y}) = R_\Omega(\mathbf{X}^T\mathbf{G}_0\mathbf{Y})$ and $\mathbf{X}^T(\mathbf{G}_0 + \Delta\mathbf{G})\mathbf{Y} = P_\mathbf{X}(\mathbf{X}^T(\mathbf{G}_0 + \Delta\mathbf{G})\mathbf{Y})P_\mathbf{Y}$, as $\mathbf{G}_0 + \Delta\mathbf{G}$ minimizes the original problem.

(2) $\mathbf{X}^T\Delta\mathbf{G}\mathbf{Y} \neq 0$, since $\mathbf{X}\mathbf{X}^T\Delta\mathbf{G}\mathbf{Y}\mathbf{Y}^T \neq 0$ for $\mathbf{X}$ and $\mathbf{Y}$ are full row rank.

(3) $\mathbf{X}^T\Delta\mathbf{G}\mathbf{Y} = P_\mathbf{X}(\mathbf{X}^T\Delta\mathbf{G}\mathbf{Y})P_\mathbf{Y}$, $R_\Omega(\mathbf{X}\Delta\mathbf{G}\mathbf{Y}) = 0$.

(4) $\|P_T(\mathbf{X}^T\Delta\mathbf{G}\mathbf{Y})\|_F \leq \zeta\|P_T(\mathbf{X}^T\Delta\mathbf{G}\mathbf{Y})\|_F \leq \zeta\|P_{T^\perp}(\mathbf{X}^T\Delta\mathbf{G}\mathbf{Y})\|_*$ since $\mathbf{X}^T\mathbf{G}_0\mathbf{Y} \neq 0$ with Condition A1.

(5) $\mathbf{U}_\perp$ and $\mathbf{V}_\perp$ are the left and right singular vectors of $P_T(\mathbf{X}^T\Delta\mathbf{G}\mathbf{Y})$, while $\mathbf{U}^T\mathbf{U}_\perp = 0$ and $\mathbf{V}^T\mathbf{V}_\perp = 0$.

Denote $\lambda = \lambda_G/\lambda_E$ for simplicity, then we can obtain that

$$\|\mathbf{E}_0 + \Delta\mathbf{E}\|_* + \lambda\|\mathbf{G}_0 + \Delta\mathbf{G}\|_1$$
$$=\|\mathbf{X}^T(\mathbf{G}_0 + \Delta\mathbf{G})\mathbf{Y}\|_* + \lambda\|\mathbf{G}_0 + \Delta\mathbf{G}\|_1$$
$$=\|\mathbf{X}^T(\mathbf{G}_0 + \Delta\mathbf{G})\mathbf{Y}\|_*\|\mathbf{U}\mathbf{V}^T + \mathbf{U}_\perp\mathbf{V}_\perp^T\| + \lambda\|\mathbf{G}_0 + \Delta\mathbf{G}\|_1$$

since the $\|\mathbf{U}\mathbf{V}^T + \mathbf{U}_\perp\mathbf{V}_\perp^T\| = 1$, where the norm is the operator norm. Then

$$\|\mathbf{X}^T(\mathbf{G}_0 + \Delta\mathbf{G})\mathbf{Y}\|_*\|\mathbf{U}\mathbf{V}^T + \mathbf{U}_\perp\mathbf{V}_\perp^T\| + \lambda\|\mathbf{G}_0 + \Delta\mathbf{G}\|_1$$
$$\geq \left\langle \mathbf{X}^T(\mathbf{G}_0 + \Delta\mathbf{G})\mathbf{Y}, \mathbf{U}\mathbf{V}^T + \mathbf{U}_\perp\mathbf{V}_\perp^T \right\rangle + \lambda\|\mathbf{G}_0 + \Delta\mathbf{G}\|_1$$
$$= \left\langle \mathbf{X}^T\mathbf{G}_0\mathbf{Y}, \mathbf{U}\mathbf{V}^T \right\rangle + \left\langle \mathbf{X}^T\mathbf{G}_0\mathbf{Y}, \mathbf{U}_\perp\mathbf{V}_\perp^T \right\rangle +$$
$$\left\langle \mathbf{X}^T\Delta\mathbf{G}\mathbf{Y}, \mathbf{U}\mathbf{V}^T + \mathbf{U}_\perp\mathbf{V}_\perp^T \right\rangle + \lambda\|\mathbf{G}_0 + \Delta\mathbf{G}\|_1$$
$$= \|\mathbf{E}_0\|_* + \left\langle \mathbf{X}^T\Delta\mathbf{G}\mathbf{Y}, \mathbf{U}\mathbf{V}^T + \mathbf{U}_\perp\mathbf{V}_\perp^T - \mathbf{H} \right\rangle + \lambda\|\mathbf{G}_0 + \Delta\mathbf{G}\|_1$$

This is obtained from the assumption A2 and matrix norm inequality. One can obtain the derivation by the norm inequality as follows,

$$\left\langle \mathbf{X}^T\Delta\mathbf{G}\mathbf{Y}, \mathbf{U}\mathbf{V}^T + \mathbf{U}_\perp\mathbf{V}_\perp^T - \mathbf{H} \right\rangle + \lambda\|\mathbf{G}_0 + \Delta\mathbf{G}\|_1$$
$$= \left\langle P_T(\mathbf{X}^T\Delta\mathbf{G}\mathbf{Y}), \mathbf{U}\mathbf{V}^T - P_T(\mathbf{H}) \right\rangle + \left\langle P_{T^\perp}(\mathbf{X}^T\Delta\mathbf{G}\mathbf{Y}), \mathbf{U}_\perp\mathbf{V}_\perp^T - P_{T^\perp}(\mathbf{H}) \right\rangle + \lambda\|\mathbf{G}_0 + \Delta\mathbf{G}\|_1$$
$$\geq \|P_{T^\perp}(\mathbf{X}^T\Delta\mathbf{G}\mathbf{Y})\|_* - \|P_T(\mathbf{X}^T\Delta\mathbf{G}\mathbf{Y})\|_F\|\mathbf{U}\mathbf{V}^T - P_T(\mathbf{H})\|_F$$
$$- \|P_{T^\perp}(\mathbf{H})\|_F\|P_{T^\perp}(\mathbf{X}^T\Delta\mathbf{G}\mathbf{Y})\|_* + \lambda\|\mathbf{G}_0 + \Delta\mathbf{G}\|_1$$

$$(1)$$

using the assumption A1 and A2 then organizing the terms, we could get

$$\|\mathbf{E}_0 + \Delta\mathbf{E}\|_* + \lambda\|\mathbf{G}_0 + \Delta\mathbf{G}\|_1$$
$$> \|\mathbf{E}_0\|_* + \|P_{T^\perp}(\mathbf{X}^T\Delta\mathbf{G}\mathbf{Y})\|_*(\frac{1}{2} - \zeta\sqrt{\frac{r}{2a}}) + \lambda\|\mathbf{G}_0 + \Delta\mathbf{G}\|_1$$
$$\geq \|\mathbf{E}_0\|_* + \|P_{T^\perp}(\mathbf{X}^T\Delta\mathbf{G}\mathbf{Y})\|_*(\frac{1}{2} - \zeta\sqrt{\frac{r}{2a}}) + \lambda\|\mathbf{G}_0\|_1 - \lambda s \qquad (2)$$
$$\geq \|\mathbf{E}_0\|_* + \lambda\|\mathbf{G}_0\|_1 + \|P_{T^\perp}(\mathbf{X}^T\Delta\mathbf{G}\mathbf{Y})\|_*(\frac{1}{2} - \zeta\sqrt{\frac{r}{2a}}) - \lambda s$$

Since $\|P_{T^\perp}(\mathbf{X}^T\Delta\mathbf{G}\mathbf{Y})\|_*(\frac{1}{2} - \zeta\sqrt{\frac{r}{2a}}) \geq 0$ which is implied from assumption A2, by observing Eq. (2), once (i) $\lambda(\|\mathbf{G}_0 + \Delta\mathbf{G}\|_1) \geq \lambda s \geq \lambda\|\mathbf{G}_0\|_1$, or (ii) $\|P_{T^\perp}(\mathbf{X}^T\Delta\mathbf{G}\mathbf{Y})\|_*(\frac{1}{2} - \zeta\sqrt{\frac{r}{2a}}) - \lambda s \geq 0$ is proved, the result could lead to a contradiction. We prove it by separating the problem into two cases.

34  If the case (i) holds, we can directly obtain from Eq. (2) that $\|\mathbf{E}_0 + \Delta\mathbf{E}\|_* + \lambda\|\mathbf{G}_0 + \Delta\mathbf{G}\|_1 \geq$
35  $\|\mathbf{E}_0\|_* + \lambda\|\mathbf{G}_0\|_1$.

36  In the contrary case of case (i), let us assume $\lambda(\|\mathbf{G}_0 + \Delta\mathbf{G}\|_1) < \lambda\|\mathbf{G}_0\|_1$. First consider the case
37  if the possible minimizers $\mathbf{G}$'s exist in the small $\varepsilon$-ball $B_\epsilon(\mathbf{G}_0)$ as a continuous neighbour of $\mathbf{G}_0$,
38  such that $\varepsilon < \min_{i,j}|\mathbf{G}_{ij}|$. Then for each $\mathbf{G}_s = \{\mathbf{G}_0 + \Delta\mathbf{G} \in B_\epsilon(\mathbf{G}_0)\}$, it satisfies $\|\mathbf{G}_s\|_1 \geq$
39  $\|\mathbf{G}_0\|_1 - ab\varepsilon$. Hence $\lambda(\|\mathbf{G}_0 + \Delta\mathbf{G}\|_1) \geq \lambda s - ab\lambda\varepsilon$. Since $\varepsilon$ is arbitrary, $\lambda(\|\mathbf{G}_0 + \Delta\mathbf{G}\|_1) \geq \lambda s$.
40  Therefore, in this case the condition (i) is satisfied, which leads to the contradiction.

Otherwise, consider the minimizers $\mathbf{G}$'s exist outside of the $\varepsilon$-ball $B_\epsilon(\mathbf{G}_0)$, which means $\mathbf{G}_0$ is an isolated minimizer. Let us assume that there exists a constant $C'$ such that for all $\Delta\mathbf{G}$,

$$\|P_{T^\perp}(\mathbf{X}^T\Delta\mathbf{G}\mathbf{Y})\|_F \geq C' > C > 0.$$

41  Here from the assumption A3 we can derive

$$\|P_{T^\perp}(\mathbf{X}^T\Delta\mathbf{G}\mathbf{Y})\|_*(\frac{1}{2} - \zeta\sqrt{\frac{r}{2a}}) - \lambda s$$

$$\geq\|P_{T^\perp}(\mathbf{X}^T\Delta\mathbf{G}\mathbf{Y})\|_F(\frac{1}{2} - \zeta\sqrt{\frac{r}{2a}}) - \lambda s$$

$$\geq C(\frac{1}{2} - \zeta\sqrt{\frac{r}{2a}}) - \lambda s$$

$$\geq 0$$

42  Thus the condition (ii) is satisfied.

43  Suppose there is no such a constant $C'$ satisfying the above condition. This implies that there exists
44  infinite minimizers and we can obtain a sub-sequence $\{\Delta\mathbf{G}_{t_k}\}_{k=1}^\infty$ satisfying

$$\lim_{k=\infty}\|P_{T^\perp}(\mathbf{X}^T\Delta\mathbf{G}_{t_k}\mathbf{Y})\|_F = 0 \tag{3}$$

45  Due to the nuclear norm inequality and the fact (4) we further infer that

$$0 \leq \lim_{k=\infty}\|P_T(\mathbf{X}^T\Delta\mathbf{G}_{t_k}\mathbf{Y})\|_F \leq \lim_{k=\infty}\|P_{T^\perp}(\mathbf{X}^T\Delta\mathbf{G}_{t_k}\mathbf{Y})\|_F = 0 \tag{4}$$

46  Combining Eq.(3) and Eq.(4) we have

$$\lim_{k=\infty}\|\mathbf{X}^T\Delta\mathbf{G}_{t_k}\mathbf{Y}\|_F = 0 \tag{5}$$

47  Eq.(5) implies that the infinite sequence $\{\mathbf{G} + \Delta\mathbf{G}_t\}_{t=N}^\infty \subset B_\epsilon(\mathbf{G}_0)$, which is contradicted to the
48  fact that no minimizers $\mathbf{G}$'s exist within the $\varepsilon$-ball $B_\epsilon(\mathbf{G}_0)$. Therefore, the above all clarify the truth
49  that the $\mathbf{E}_0 + \Delta\mathbf{E}$ and $\mathbf{G}_0 + \Delta\mathbf{G}$ are not the minimizer for our optimization problem. ∎

## 50  1.1  A1 holds with high probability

51  In this subsection we prove that Lemma 2 holds with some certain probability. Lemma 2 roots from
52  combining the results from Lemma 5 and Lemma 6, which upper-bounds $\|P_T(\mathbf{M})\|_F$ and lower-
53  bounds $\|P_{T^\perp}(\mathbf{M})\|_F$ and clarify the inequality between them. Lemma 3 and 4 are cited from [7] to
54  facilitate the proof.

55  Let's first illustrate Lemma 2 as below.

**Lemma 2** *With a certain probability at least $1 - 4N^{-p+1}$, for any $\mathbf{M} \neq 0$, $\mathbf{M} \in \mathbb{R}^{m \times n}$ satisfying $R_\Omega(\mathbf{M}) = 0$ and $\mathbf{M} = P_\mathbf{X}\mathbf{M}P_\mathbf{Y}$ we have*

$$\|P_T(\mathbf{M})\|_F \leq \zeta\|P_{T^\perp}(\mathbf{M})\|_F,$$

56  *where $\zeta$ is the same as in Lemma 1, if $T_0 \leq |\mathbf{\Omega}| \leq T_1$.*

**Proof.**  Since $R_\Omega(\mathbf{M}) = 0$ and $\mathbf{M} = P_\mathbf{X}\mathbf{M}P_\mathbf{Y}$, we have $R_\Omega P_T(\mathbf{M}) = -R_\Omega P_{T^\perp}(\mathbf{M})$. Then we could attain
$$\frac{mn}{|\mathbf{\Omega}|}\langle\mathbf{M}, P_T R_\Omega P_T(\mathbf{M})\rangle = \frac{mn}{|\mathbf{\Omega}|}\langle\mathbf{M}, P_{T^\perp}R_\Omega P_{T^\perp}(\mathbf{M})\rangle$$

57 First, according to Lemma 5 and Lemma 6, with a probability at least $1 - 4N^{-p+1}$, we have

$$
\begin{aligned}
\frac{1}{2}\|P_T(\mathbf{M})\|_F^2 &\le \frac{mn}{|\mathbf{\Omega}|}\left\langle \mathbf{M}, P_T R_\Omega P_T(\mathbf{M})\right\rangle \\
&\le \frac{16\sigma^2 p\mu^2(ab+r^2)\log N}{3|\mathbf{\Omega}|}\|P_{T^\perp}(\mathbf{M})\|_F^2 \\
&\le \frac{16\sigma^2 p\mu^2(ab+r^2)\log N}{3T_0}\|P_{T^\perp}(\mathbf{M})\|_F^2 \\
&= \frac{1}{2}\|P_{T^\perp}(\mathbf{M})\|_F^2
\end{aligned}
$$

finally we have

$$
\frac{1}{2}\|P_T(\mathbf{M})\|_F \le \frac{1}{\sqrt{2}}\|P_{T^\perp}(\mathbf{M})\|_F
$$

58 while $r \le a$, we have $\frac{1}{\sqrt{2}} \le \sqrt{\frac{a}{2r}}$, so the lemma proved. ∎

59 Before we prove Lemma 5 and Lemma 6, which we which upper-bounds $\|P_T(\mathbf{M})\|_F$ and lower-
60 bounds $\|P_{T^\perp}(\mathbf{M})\|_F$ on certain conditions, we first need to illustrate Lemma 3,Lemma 4 derived
61 from the Bernstein Inequality [5].

**Lemma 3** *Let $\mathbf{X}_1, ..., \mathbf{X}_L$ be independent zero-mean random matrices of dimension $d_1 \times d_2$. Suppose $\rho_k^2 \ge \max\{\|\mathbb{E}[\mathbf{X}_k\mathbf{X}_k^T]\|, \|\mathbf{X}_k^T\mathbf{X}_k\|\}$ and $\|\mathbf{X}_k\| \le M$ almost surely form all k. If we assume*

$$
M^2 \log\frac{d_1+d_2}{\xi} \le \frac{3}{8}\sum \rho_k^2,
$$

*then with a certain probability at least $1 - \xi$, we have,*

$$
\|\sum_{k=1}^{L}\mathbf{X}_k\| \le \sqrt{\frac{8}{3}\ln\frac{d_1+d_2}{\xi}\sum_{k=1}^{L}\rho_k^2}.
$$

62 We can also give Lemma 4 which can be derived from Lemma 3 as below;

**Lemma 4** *Let $\mathbf{X}_1, ..., \mathbf{X}_L$ be independent zero-mean random matrices of dimension $d_1 \times d_2$. Suppose $\rho_k^2 \ge \max\{\|\mathbb{E}[\mathbf{X}_k\mathbf{X}_k^T]\|, \|\mathbf{X}_k^T\mathbf{X}_k\|\}$ and $\|\mathbf{X}_k\| \le M$ almost surely form all k. If we assume*

$$
M^2 \log\frac{d_1+d_2}{\xi} \le \frac{3}{8}\sum \rho_k^2,
$$

*then with a certain probability at least $1 - \xi$, we have,*

$$
\|\sum_{k=1}^{L}\mathbf{X}_k\| \le \frac{8}{3}M\log\frac{d_1+d_2}{\xi}.
$$

63 next we will bound $\|P_T - \frac{mn}{|\Omega|}P_T R_\Omega P_T\|$ by using Lemma 3 and Lemma 4.

**Lemma 5** *With a certain probability at least $1 - 2N^{-p+1}$, we have*

$$
\left\|P_T - \frac{mn}{|\mathbf{\Omega}|}P_T R_\Omega P_T\right\| \le \sqrt{\frac{8p\mu^2 r(a+b)\log N}{3|\mathbf{\Omega}|}}
$$

*if $|\Omega| \ge \frac{8p}{3}\mu^2 r(a+b)\log N$ and therefore, for any $\mathbf{M} \in \mathbb{R}^{m\times n}$,*

$$
\frac{mn}{|\mathbf{\Omega}|}\left\langle \mathbf{M}, P_T R_\Omega P_T(\mathbf{M})\right\rangle \ge \frac{1}{2}\|P_T(\mathbf{M})\|_F^2
$$

64 *if $|\mathbf{\Omega}| \ge T_0$.*

**Proof.** For any $\mathbf{M} \in \mathbb{R}^{m \times n}$, we have

$$P_T R_\Omega P_T(\mathbf{M}) = \sum_{(i,j) \in \Omega} \left\langle P_T(\mathbf{M}), \mathbf{e}_i \mathbf{e}_j^T \right\rangle P_T(\mathbf{e}_i \mathbf{e}_j^T) = \sum_{(i,j) \in \Omega} \left\langle \mathbf{M}, P_T(\mathbf{e}_i \mathbf{e}_j^T) \right\rangle P_T(\mathbf{e}_i \mathbf{e}_j^T).$$

For any $i \in [m]$ and $j \in [n]$, define linear operator $T_{i,j}$ as

$$T_{i,j}(\mathbf{M}) = \left\langle \mathbf{M}, P_T(\mathbf{e}_i \mathbf{e}_j^T) \right\rangle P_T(\mathbf{e}_i \mathbf{e}_j^T) = P_T R_{(i,j)} P_T(\mathbf{M}),$$

where $R_{(i,j)}(\mathbf{M}) = \mathbf{e}_i \mathbf{e}_j^T \mathbf{M}_{i,j}$. So that

$$P_T R_\Omega P_T(\mathbf{M}) = \sum_{(i,j) \in \Omega} P_T R_{(i,j)} P_T(\mathbf{M}) = \sum_{(i,j) \in \Omega} T_{i,j}(\mathbf{M}).$$

65  To implement Lemma 3, we need to give $M$ and the corresponding $\rho^2$. Since $\|P_T - \frac{mn}{|\mathbf{\Omega}|} P_T R_\Omega P_T\|$
66  can be viewed as the spectral norm of $|\mathbf{\Omega}|$ independent zero-mean random variables $\frac{1}{|\mathbf{\Omega}|} P_T - \frac{mn}{|\mathbf{\Omega}|} T_{i,j}$,
67  then we have

$$\|\frac{1}{|\mathbf{\Omega}|} P_T - \frac{mn}{|\mathbf{\Omega}|} T_{i,j}\| \leq \max\{\|\frac{1}{|\mathbf{\Omega}|} P_T\|, \|\frac{mn}{|\mathbf{\Omega}|} T_{i,j}\|\}$$

$$= \max\{\|\frac{1}{|\mathbf{\Omega}|} P_T\|, \frac{mn}{|\mathbf{\Omega}|} \arg \max_{\|\mathbf{M}\|_F = 1} \| \left\langle \mathbf{M}, P_T(\mathbf{e}_i \mathbf{e}_j^T) \right\rangle P_T(\mathbf{e}_i \mathbf{e}_j^T)\|_F\}$$

$$= \max\{\|\frac{1}{|\mathbf{\Omega}|} P_T\|, \frac{mn}{|\mathbf{\Omega}|} \arg \max_{\|\mathbf{M}\|_F = 1} \left\langle \mathbf{M}, P_T(\mathbf{e}_i \mathbf{e}_j^T) \right\rangle \|P_T(\mathbf{e}_i \mathbf{e}_j^T)\|_F\}$$

$$= \max\{\|\frac{1}{|\mathbf{\Omega}|} P_T\|, \frac{mn}{|\mathbf{\Omega}|} \|P_T(\mathbf{e}_i \mathbf{e}_j^T)\|_F\}$$

68  To bound $\|P_T(\mathbf{e}_i \mathbf{e}_j^T)\|_F$, we get

$$\|P_T(\mathbf{e}_i \mathbf{e}_j^T)\|_F = \left\langle P_T(\mathbf{e}_i \mathbf{e}_j^T), \mathbf{e}_i \mathbf{e}_j^T \right\rangle$$

$$= \left\langle P_\mathbf{X}(\mathbf{e}_i \mathbf{e}_j^T) P_\mathbf{V}, \mathbf{e}_i \mathbf{e}_j^T \right\rangle + \left\langle P_\mathbf{U}(\mathbf{e}_i \mathbf{e}_j^T) P_\mathbf{Y}, \mathbf{e}_i \mathbf{e}_j^T \right\rangle - \left\langle P_\mathbf{U}(\mathbf{e}_i \mathbf{e}_j^T) P_\mathbf{V}, \mathbf{e}_i \mathbf{e}_j^T \right\rangle$$

$$= \|P_\mathbf{X}(\mathbf{e}_i \mathbf{e}_j^T) P_\mathbf{V}\|_F + \|P_\mathbf{U}(\mathbf{e}_i \mathbf{e}_j^T) P_\mathbf{Y}\|_F - \|P_\mathbf{U}(\mathbf{e}_i \mathbf{e}_j^T) P_\mathbf{V}\|_F$$

$$\leq \|P_\mathbf{X} \mathbf{e}_i\|_F \|P_\mathbf{V} \mathbf{e}_j\|_F + \|P_\mathbf{U} \mathbf{e}_i\|_F \|P_\mathbf{Y} \mathbf{e}_j\|_F$$

$$\leq \|\mathbf{X}^T \mathbf{V}_\mathbf{X} \mathbf{\Sigma}_\mathbf{X}^{-2} \mathbf{V}_\mathbf{X}^T\|_F \frac{a\mu_\mathbf{XY}}{m} \frac{r\mu_0}{n} + \|\mathbf{Y}^T \mathbf{V}_\mathbf{Y} \mathbf{\Sigma}_\mathbf{Y}^{-2} \mathbf{V}_\mathbf{Y}^T\|_F \frac{r\mu_0}{m} \frac{b\mu_\mathbf{XY}}{n}$$

$$\leq \|\mathbf{\Sigma}_\mathbf{X}^{-1}\|_* \frac{a\mu_\mathbf{XY}}{m} \frac{r\mu_0}{n} + \|\mathbf{\Sigma}_\mathbf{Y}^{-1}\|_* \frac{r\mu_0}{m} \frac{b\mu_\mathbf{XY}}{n}$$

$$\leq \sigma \frac{r\mu_0 \mu_\mathbf{XY}(a+b)}{mn} \leq \frac{\sigma r\mu^2(a+b)}{mn}.$$

69  Therefore

$$\|\frac{1}{|\mathbf{\Omega}|} P_T - \frac{mn}{|\mathbf{\Omega}|} T_{i,j}\| \leq \max\{\|\frac{1}{|\mathbf{\Omega}|} P_T\|, \frac{mn}{|\mathbf{\Omega}|} \|P_T(\mathbf{e}_i \mathbf{e}_j^T)\|_F\}$$

$$\leq \max\{\|\frac{1}{|\mathbf{\Omega}|} P_T\|, \frac{\sigma r\mu^2(a+b)}{mn}\} = \max\{\frac{1}{|\mathbf{\Omega}|}, \frac{\sigma r\mu^2(a+b)}{mn}\}$$

$$= \max\{\frac{1}{|\mathbf{\Omega}|}, \frac{\sigma r\mu^2(a+b)}{mn}\} = \frac{\sigma r\mu^2(a+b)}{mn} = M$$

70  Since that $\frac{1}{|\mathbf{\Omega}|}\mathbb{E}[P_T R_\Omega P_T(\mathbf{M})] = \frac{1}{mn}P_T(\mathbf{M})$, the corresponding $\rho_{i,j}^2$ can be calculated as

$$\begin{aligned}
\rho_{i,j}^2 &= \|\mathbb{E}[(\frac{1}{|\mathbf{\Omega}|}P_T - \frac{mn}{|\mathbf{\Omega}|}T_{i,j})^T(\frac{1}{|\mathbf{\Omega}|}P_T - \frac{mn}{|\mathbf{\Omega}|}T_{i,j})]\| \\
&= \|\mathbb{E}[\frac{1}{|\mathbf{\Omega}|^2}P_T P_T + \frac{m^2 n^2}{|\mathbf{\Omega}|^2}T_{i,j}T_{i,j} - \frac{2mn}{|\mathbf{\Omega}|^2}P_T T_{i,j}]\| \\
&= \|\frac{1}{|\mathbf{\Omega}|^2}P_T + \frac{m^2 n^2}{|\mathbf{\Omega}|^2}\mathbb{E}[T_{i,j}T_{i,j}] - \frac{2mn}{|\mathbf{\Omega}|^2}P_T \mathbb{E}[T_{i,j}]\| \\
&= \|\frac{1}{|\mathbf{\Omega}|^2}P_T + \frac{m^2 n^2}{|\mathbf{\Omega}|^2}\mathbb{E}[T_{i,j}T_{i,j}] - \frac{2mn}{|\mathbf{\Omega}|^2}P_T \frac{1}{mn}P_T\| \\
&= \|\frac{m^2 n^2}{|\mathbf{\Omega}|^2}\mathbb{E}[T_{i,j}T_{i,j}] - \frac{1}{|\mathbf{\Omega}|^2}P_T\| \le \max\{\frac{m^2 n^2}{|\mathbf{\Omega}|^2}\mathbb{E}[T_{i,j}T_{i,j}], \frac{1}{|\mathbf{\Omega}|^2}P_T\} \\
&\le \max\{\frac{m^2 n^2}{|\mathbf{\Omega}|^2}\mathbb{E}[\|P_T(\mathbf{e}_i\mathbf{e}_j^T)\|_F\|T_{i,j}\|], \frac{1}{|\mathbf{\Omega}|^2}\} \\
&\le \max\{\frac{m^2 n^2}{|\mathbf{\Omega}|^2}\frac{\sigma r\mu^2(a+b)}{mn}\frac{1}{mn}\|P_T\|], \frac{1}{|\mathbf{\Omega}|^2}\} \\
&= \frac{\sigma r\mu^2(a+b)}{|\mathbf{\Omega}|^2}
\end{aligned}$$

71  By Lemma 4, let $M = \frac{\sigma r\mu^2(a+b)}{mn}$ and $\rho^2 = \frac{\sigma r\mu^2(a+b)}{|\mathbf{\Omega}|^2}$, we conclude with a certain probability
72  $1 - 2N^{-p+1}$,

$$\|P_T - \frac{mn}{|\mathbf{\Omega}|}P_T R_\Omega P_T\| \le \sqrt{\frac{8}{3}\log\frac{m+n}{2N^{-p+1}}\frac{\sigma r\mu^2(a+b)}{|\mathbf{\Omega}|}} \le \sqrt{\frac{8\sigma pr\mu^2(a+b)\log N}{3|\mathbf{\Omega}|}}$$

which also should satisfy the condition that

$$\frac{\sigma^2 r^2\mu^4(a+b)^2}{|\mathbf{\Omega}|^2}\log\frac{m+n}{2N^{-p+1}} \le \frac{3}{8}\frac{\sigma r\mu^2(a+b)}{|\mathbf{\Omega}|}$$

which means

$$|\mathbf{\Omega}| \ge \frac{8\sigma pr\mu^2(a+b)\log N}{3}.$$

Moreover, if $|\mathbf{\Omega}| \ge T_0 \ge \frac{32\sigma pr\mu^2(a+b)\log N}{3}$, then

$$\left\|P_T - \frac{mn}{|\mathbf{\Omega}|}P_T R_\Omega P_T\right\| \le \sqrt{\frac{8\sigma p\mu^2 r(a+b)\log\tau}{3|\mathbf{\Omega}|}} \le \frac{1}{2},$$

By utilizing the property of matrix norm, we have

$$\left\langle \mathbf{M}, P_T(\mathbf{M}) - \frac{mn}{|\mathbf{\Omega}|}P_T R_\Omega P_T(\mathbf{M})\right\rangle \le \frac{1}{2}\|P_T(\mathbf{M})\|_F^2$$

So that

$$\langle \mathbf{M}, P_T(\mathbf{M})\rangle - \frac{1}{2}\|P_T(\mathbf{M})\|_F \le \left\langle \mathbf{M}, \frac{mn}{|\mathbf{\Omega}|}P_T R_\Omega P_T(\mathbf{M})\right\rangle,$$

which we can easily derive

$$\frac{1}{2}\|P_T(\mathbf{M})\|_F^2 \le \frac{mn}{|\mathbf{\Omega}|}P_T R_\Omega P_T(\mathbf{M}).$$

73  ∎

74  Following the similarly outline of the proof as Lemma 5, we can prove the following Lemma 6.

**Lemma 6** *With a certain probability at least* $1 - 2N^{-p+1}$, *we have*

$$\left\| P_{T^\perp} - \frac{mn}{|\mathbf{\Omega}|} P_{T^\perp} R_\Omega P_{T^\perp} \right\| \leq \frac{8\sigma^2 p\mu^2(ab + r^2)\log N}{3|\mathbf{\Omega}|}$$

*if* $|\Omega| \geq T_0 = \frac{8}{3}\sigma\mu^2 pr(a + b)\log N$ *and therefore, for any* $\mathbf{M} \in \mathbb{R}^{m \times n}$,

$$\frac{mn}{|\mathbf{\Omega}|} \langle \mathbf{M}, P_{T^\perp} R_\Omega P_{T^\perp}(\mathbf{M}) \rangle \leq \frac{16\sigma^2 p\mu^2(ab + r^2)\log N}{3|\mathbf{\Omega}|} \|P_{T^\perp}(\mathbf{M})\|_F^2$$

75  Then based on Lemma 5 and 6, we can prove that **A1** holds with a certain high probability.

## 1.2  A2 holds with high probability

77  In this subsection we aim to investigate the condition when **A2** holds with the high probabili-
78  ty. Like the similar approach we propose above, we also need to bound the following two terms
79  $\frac{mn}{|\mathbf{\Omega}|}\|P_{T^\perp} R_\Omega P_T(\mathbf{H})\|_F$ and $\|P_T(\mathbf{H}) - \frac{mn}{|\mathbf{\Omega}|} P_{T^\perp} R_\Omega P_T(\mathbf{H})\|_\infty$ in Lemma 7 and 8 respectively where
80  $\|\cdot\|_\infty$ is the maximum entry of a matrix.

**Lemma 7** *For a fixed* $\mathbf{H} \in \mathbb{R}^{m \times n}$, *with a probability* $1 - 2N^{-p+1}$, *we have*

$$\frac{mn}{|\Omega|}\|P_{T^\perp} R_\Omega P_T(\mathbf{H})\| \leq \|P_T(\mathbf{H})\|_\infty \sqrt{\frac{8\sigma pmn\mu a\log N}{3|\mathbf{\Omega}|}},$$

81  *if* $|\mathbf{\Omega}| \geq T_0$.

**Proof.**  We write

$$P_{T^\perp} R_\Omega P_T(F) = \sum_{(i,j) \in \mathbf{\Omega}} \langle \mathbf{H}, P_T(\mathbf{e}_i \mathbf{e}_j^T) \rangle P_{T^\perp}(\mathbf{e}_i \mathbf{e}_j^T) = \sum_{(i,j) \in \mathbf{\Omega}} T_{i,j},$$

where $T_{i,j}(\mathbf{H}) = \langle \mathbf{H}, P_T(\mathbf{e}_i \mathbf{e}_j^T) \rangle P_{T^\perp}(\mathbf{e}_i \mathbf{e}_j^T)$. Evidently,

$$\mathbb{E}[P_{T^\perp} R_\Omega P_T(\mathbf{H})] = 0.$$

82  To use Lemma 3, we compute $M$ and $\rho^2$ as,

$$
\begin{aligned}
M &= \max_{i \in [m] j \in [n]} \|T_{i,j}\| \\
&\leq \max_{i \in [m] j \in [n]} \max_{\|F\|_F = 1} \|\langle \mathbf{H}, P_T(\mathbf{e}_i \mathbf{e}_j^T) \rangle P_{T^\perp}(\mathbf{e}_i \mathbf{e}_j^T)\|_F \\
&\leq \max_{i \in [m] j \in [n]} \langle \mathbf{H}, P_T(\mathbf{e}_i \mathbf{e}_j^T) \rangle P_{T^\perp}(\mathbf{e}_i \mathbf{e}_j^T) \\
&\leq \|P_T(\mathbf{H})\|_\infty \max_{i \in [m] j \in [n]} \|P_{T^\perp}(\mathbf{e}_i \mathbf{e}_j^T)\|_F \\
&\leq \|P_T(\mathbf{H})\|_\infty \sqrt{\frac{\mu^2 \sigma^2 (ab + r^2)}{mn}}
\end{aligned}
$$

83  and

$$
\begin{aligned}
\rho_{i,j}^2 &= \max\{\|\mathbb{E}[T_{i,j}, T_{i,j}^T]\|, \|\mathbb{E}[T_{i,j}^T, T_{i,j}]\|\} \\
&= \|P_T(\mathbf{H})\|_\infty^2 \max\{\|\mathbb{E}[P_{T^\perp}(\mathbf{e}_i \mathbf{e}_j^T)^T P_{T^\perp}(\mathbf{e}_i \mathbf{e}_j^T)]\|, \|\mathbb{E}[P_{T^\perp}(\mathbf{e}_i \mathbf{e}_j^T) P_{T^\perp}(\mathbf{e}_i \mathbf{e}_j^T)^T]\|\} \\
&= \|P_T(\mathbf{H})\|_\infty^2 \max\{\|\mathbb{E}[P_{Y^\perp}\mathbf{e}_j \mathbf{e}_i^T P_{X^\perp}\mathbf{e}_i \mathbf{e}_j^T P_{Y^\perp}]\|, \|\mathbb{E}[P_{X^\perp}\mathbf{e}_i \mathbf{e}_j^T P_{Y^\perp}\mathbf{e}_j \mathbf{e}_i^T P_{X^\perp}]\|\} \\
&\leq \|P_T(\mathbf{H})\|_\infty^2 \max\{\frac{\sigma\mu_{\mathbf{XY}} a}{m}\|\mathbb{E}[P_{Y^\perp}\mathbf{e}_j \mathbf{e}_j^T P_{Y^\perp}]\|, \frac{\sigma\mu_{\mathbf{XY}} b}{n}\|\mathbb{E}[P_{X^\perp}\mathbf{e}_i \mathbf{e}_i^T P_{X^\perp}]\|,\} \\
&\leq \|P_T(\mathbf{H})\|_\infty^2 \sigma \max\{\frac{\mu_{\mathbf{XY}} a}{m}\|P_{Y^\perp}\mathbb{E}[\mathbf{e}_j \mathbf{e}_j^T]P_{Y^\perp}\|, \frac{\mu_{\mathbf{XY}} b}{n}\|P_{X^\perp}\mathbb{E}[\mathbf{e}_i \mathbf{e}_i^T]P_{X^\perp}\|,\} \\
&\leq \|P_T(\mathbf{H})\|_\infty^2 \sigma \max\{\frac{\mu_{\mathbf{XY}} a}{mn}\|P_{Y^\perp}P_{Y^\perp}\|, \frac{\mu_{\mathbf{XY}} b}{mn}\|P_{X^\perp}P_{X^\perp}\|,\} \\
&\leq \|P_T(\mathbf{H})\|_\infty^2 \frac{\sigma\mu_{\mathbf{XY}} \max\{a, b\}}{mn}.
\end{aligned}
$$

To prove simply without loss of the generality, we assume $b \leq a$, so we can get

$$\rho_{i,j}^2 \leq \|P_T(\mathbf{H})\|_\infty^2 \frac{\sigma\mu_{\mathbf{XY}}a}{mn} \leq \|P_T(\mathbf{H})\|_\infty^2 \frac{\sigma\mu a}{mn}$$

By Lemma 3, we have, if

$$\|P_T(\mathbf{H})\|_\infty^2 \frac{\sigma^2\mu^2(ab+r^2)}{mn} \log \frac{2N}{2N^{-p+1}} \leq \frac{3}{8}\|P_T(\mathbf{H})\|_\infty^2 \frac{\sigma\mu a|\mathbf{\Omega}|}{mn}$$

that is

$$\frac{8\sigma\mu(ab+r^2)p\log N}{3a} \leq |\mathbf{\Omega}|,$$

therefore, with a probability of $1 - 2N^{-p+1}$,

$$\begin{aligned}
\frac{mn}{|\mathbf{\Omega}|} \leq & \frac{mn}{|\mathbf{\Omega}|}\|P_T(\mathbf{H})\|_\infty \sqrt{\frac{8\sigma p\rho^2|\mathbf{\Omega}|\log N}{3}} \\
\leq & \frac{mn}{|\mathbf{\Omega}|}\|P_T(\mathbf{H})\|_\infty \sqrt{\frac{8\sigma p\mu a|\mathbf{\Omega}|\log N}{3mn}} \\
= & \|P_T(\mathbf{H})\|_\infty \sqrt{\frac{8\sigma mnp\mu a\log N}{3|\mathbf{\Omega}|}}
\end{aligned}$$

by Lemma 2, we need to use the condition $|\mathbf{\Omega}| \geq T_0$, and then

$$|\mathbf{\Omega}| \geq T_0 \geq \frac{32\sigma pr\mu^2(a+b)\log N}{3} \geq \frac{8\sigma p(ab+r^2)\mu\log N}{3a}$$

which is because $\mu \geq 1$, $a \geq b$, and $a \gg r$. Then under the condition $|\mathbf{\Omega}| \geq T_0$, we complete the proof. ∎

**Lemma 8** *For a fixed* $\mathbf{H} \in \mathbb{R}^{m \times n}$, *with a probability* $1 - 2N^{-p+2}$, *we have*

$$\| \left( P_T - \frac{mn}{|\mathbf{\Omega}|P_T R_\Omega P_T} \right)(\mathbf{H})\|_\infty \leq \sqrt{\frac{8\sigma pr\mu^2(a+b)\log N}{3|\mathbf{\Omega}|}}\|P_T(\mathbf{H})\|_\infty$$

*and therefore if* $|\mathbf{\Omega}| \leq T_0$,

$$\| \left( P_T - \frac{mn}{|\mathbf{\Omega}|P_T R_\Omega P_T} \right)(\mathbf{H})\|_\infty \leq \frac{1}{2}\|P_T(\mathbf{F})\|_\infty$$

**Proof.** For each matrix index $(a, b)$, sample $(i, j)$ uniformly at random to define the random variable $\eta_{a,b} = [mnP_T R_\Omega P_T(\mathbf{H}) - P_T(\mathbf{H})]$ We have

$$\mathbb{E}[\eta_{a,b}] = 0,$$

$$\eta_{a,b} \leq \|P_T R_{i,j} P_T - P_T\|\|P_T(\mathbf{H})\|_\infty \leq r\sigma\mu^2(a+b)\|P_T(\mathbf{H})\|_\infty$$

and

$$\begin{aligned}
\mathbb{E}[\eta_{a,b}^2] = & \mathbb{E}[([mnP_T R_{i,j} P_T(\mathbf{H}) - P_T(\mathbf{H})]_{a,b})^2] \\
= & \mathbb{E}[([m^2n^2 P_T R_{i,j} P_T(\mathbf{H})]_{a,b})^2] + ([P_T(\mathbf{H})]_{a,b})^2 - 2mn\mathbb{E}[([P_T R_{i,j} P_T(\mathbf{H})]_{a,b}[P_T(\mathbf{H})]_{a,b})^2] \\
= & m^2n^2\mathbb{E}[([P_T R_{i,j} P_T(\mathbf{H})]_{a,b})^2] - ([P_T(\mathbf{H})]_{a,b})^2 \\
= & m^2n^2\mathbb{E}[(\langle \mathbf{e}_a\mathbf{e}_b^T, P_T(\mathbf{e}_i\mathbf{e}_j^T)\rangle \langle \mathbf{H}, P_T(\mathbf{e}_i\mathbf{e}_j^T)\rangle)^2] - ([P_T(\mathbf{H})]_{a,b})^2 \\
= & mn\|P_T(\mathbf{H})\|_F^2\|P_T(\mathbf{e}_a\mathbf{e}_b)\|_F^2 - ([P_T(\mathbf{H})]_{a,b})^2 \\
\leq & \|P_T(\mathbf{H})\|_\infty^2 r\sigma\mu^2(a+b)
\end{aligned}$$

Using the standard Bernstein Inequality, we have

$$\mathbb{P}\left[|[mnP_T R_\Omega P_T(\mathbf{H}) - |\mathbf{\Omega}|P_T(\mathbf{H})]_{a,b}| > \sqrt{\frac{8|\mathbf{\Omega}|\|P_T(\mathbf{H})\|_\infty^2 r\sigma\mu^2(a+b)\log\frac{2}{2N^{-p}}}{3}}\right] \leq 2N^{-p}$$

Take the union bound, we have , with a probability of $1 - 2N^{-p+2}$

$$\|\frac{mn}{|\boldsymbol{\Omega}|}P_T R_\Omega P_T(\mathbf{H}) - P_T(\mathbf{H})\|_\infty \leq \sqrt{\frac{9\sigma r p\mu^2(a+b)\ln N}{3|\boldsymbol{\Omega}|}}\|P_T(\mathbf{H})\|_\infty$$

If $|\boldsymbol{\Omega}| \geq T_0$, we have

$$\|\frac{mn}{|\boldsymbol{\Omega}|}P_T R_\Omega P_T(\mathbf{H})\|_\infty \leq \frac{1}{2}\|P_T(\mathbf{H})\|_\infty$$

88  ∎

Next we need to verify that there exists a matrix $\mathbf{H}$ that satisfies the conditions in assumption A2, we follow the idea in [7] and construct $F$ as follows. We generate a sequence of $\mathbf{H}_t, t = 1, ..., q$ as follows

$$\mathbf{H}_t = \frac{mn}{T_0}\sum_{i=1}^{t} R_{\Omega_i}(\mathbf{W}_i),$$

where $\mathbf{W}_1 = \mathbf{U}\mathbf{V}^T$ and $\mathbf{W}_{t+1}$ is defined as

$$\mathbf{W}_{t+1} = P_T(\mathbf{U}\mathbf{V}^T - \mathbf{H}_t) = (P_T - \frac{mn}{T_0}P_T R_{\Omega_t} P_T)(\mathbf{W}_t)$$

89  We randomly select $qT_0$ entries from $\Omega$ and partition the selected entries into $q$ subsets as
90  $\Omega_1, ..., \Omega_q$ with equal sizes, with $|\Omega_i| = T_0, \ , i = 1, ..., q$. Thus we have $\mathbf{H} = \mathbf{H}_q$ and $\mathbf{H} = R_\Omega(\mathbf{H})$.

91  Now we are ready to show that $\mathbf{H}$ satisfies the other two properties in assumption A2.

**Lemma 9** *With a probability of $1 - 2qN^{-p+1}$, it is satisfied that*

$$\|P_T(\mathbf{H})\| \leq \sqrt{\frac{r}{2a}}$$

92  *if $q \geq q_0$*

**Lemma 10** *With a probability of $1 - 2qN^{-p+1} - 2qN^{-p+2}$, it is satisfied that*

$$\|P_{T^\perp}(\mathbf{H})\| \leq \frac{1}{2}$$

93  *if $q \geq q_0$*

**Proof.** Because of Lemma 8 we have

$$\|\mathbf{H}_{t+1}\|_\infty = \|(P_T - \frac{mn}{T_0}P_T R_\Omega P_T)\mathbf{H}_t\|_\infty \leq \frac{1}{2}\|\mathbf{H}_t\|_\infty.$$

94  To bound $\|P_{T^\perp}(\mathbf{H})\|$, we have

$$\|P_{T^\perp}(\mathbf{H})\| \leq \sum_{i=1}^{q} \frac{mn}{T_0}\|P_{T^\perp} R_{\Omega_i} P_T(\mathbf{H}_i)\|$$

$$\leq \alpha \sum_{i=1}^{q}\|\mathbf{H}_i\|_\infty \leq \alpha\|\mathbf{H}_1\|_\infty \sum_{i=1}^{q}\frac{1}{2^{i-1}}$$

$$= 2\alpha\|\mathbf{H}_1\|_\infty \leq 2\sqrt{\frac{8\sigma pmn\mu a\log N}{3|\boldsymbol{\Omega}|}}\sqrt{\frac{\mu_1 r}{mn}}$$

$$\leq 2\sqrt{\frac{8\sigma pmn\mu a\log N}{3|\boldsymbol{\Omega}|}}$$

So when $|\boldsymbol{\Omega}| \geq \frac{128\sigma\mu_1 rp\mu a\log N}{3}$, it could be guaranteed that $\|P_{T^\perp}(\mathbf{H})\| \leq \frac{1}{2}$ when

$$|\boldsymbol{\Omega}| \geq \frac{128p\sigma\mu\mu_{\mathbf{XY}}r(a+b)\log N}{3} = T_0.$$

95  ∎

## 2  $\epsilon$-Recovery Sampling Complexity

Consider the optimization problem below that if the perfect feature matrices $\mathbf{X}$ and $\mathbf{Y}$ are corrupted by $\Delta\mathbf{X}$ and $\Delta\mathbf{Y}$ and bounded by a constant $\|\Delta\mathbf{X}\|_F \leq s_1$ and $\|\Delta\mathbf{Y}\|_F \leq s_2$, so that we investigate the following relaxed optimization problem:

$$\min_{\mathbf{G}} \|R_\Omega((\mathbf{X}+\Delta\mathbf{X})^T\mathbf{G}(\mathbf{Y}+\Delta\mathbf{Y}) - \mathbf{F})\|_F^2$$
$$\text{subject to } \mathbf{E} - \mathbf{X}^T\mathbf{G}\mathbf{Y} \in B(0,\phi), \tag{6}$$
$$\text{subject to } \|\mathbf{G}\|_1 \leq \alpha, \quad \|\mathbf{E}\|_* \leq \gamma.$$

where $B(0,\phi) \subset \mathbb{R}^{m\times n}$ is a ball with the radius of $\phi$ and center at $0$.

The matrix $\mathbf{F}_{ij}$ is assumed to be observed partially i.i.d. from an index set $\{(i_\alpha, j_\alpha)\}_{\alpha=1}^m$ with unknown distribution.

We denote $\Theta = \{(\mathbf{G},\mathbf{E}) \mid \|\mathbf{G}\|_1 \leq \beta, \|\mathbf{E}\|_* \leq \gamma, \mathbf{E} = \mathbf{X}^T\mathbf{G}\mathbf{Y}\}$ as the feasible solution set, and $\theta = (\mathbf{G},\mathbf{E}) \in \Theta$ as any feasible solution. Let $F_\theta(i,j) = \mathbf{x}_i^T\mathbf{G}\mathbf{y}_j$ be the estimation function for $\mathbf{F}_{ij}$ with $\theta$ as the parameters, and $F_\Theta = \{f_\theta \mid \theta \in \Theta\}$ be the set of feasible functions. Denote the loss function as $l$ where $l(f_\theta(i,j), \mathbf{F}_{ij}) = R_\Omega(\mathbf{X}^T\mathbf{G}\mathbf{Y} - \mathbf{F})_{i,j}^2$. Then, we introduce two "$l$-risk" quantities: the expected $l$-risk

$$\mathcal{R}_l(f) = \mathbb{E}_{(i,j)}[l(f_\theta(i,j), \mathbf{F}_{ij})],$$

and the empirical $l$-risk

$$\hat{\mathcal{R}}_l(f) = \frac{1}{s}\sum_{(i,j)}[l(f_\theta(i,j), \mathbf{F}_{ij})].$$

In this notation, our model is to solve for $\theta$ that parameterizes $f^* = \arg\min_{f\in F_\Theta}\hat{\mathcal{R}}_l(f)$, and it is sufficient to show that the recovery can be attained if $\hat{\mathcal{R}}_l(f^*)$ approaches to zero. Next we implement Rademacher complexity, a learning theoretic tool to measure the complexity of a function class. Then we will derive the sampling rate. To begin with, we cite the following Lemma [1] to bound the expected risk.

**Lemma 11** *(Bound on Expected risk). Let $l$ be a loss function with Lipschitz constant $L_l$ in the compact domain respect to its first argument bounded by $B$, and $p$ be a constant where $0 < p < 1$. Let $\mathfrak{R}(F_\Theta)$ be the Rademacher complexity of the function class $F_\Theta$ defined as:*

$$\mathfrak{R}(F_\Theta) = \mathbb{E}[\sup_{f\in F_\Theta} \frac{1}{s}\sum_{t=1}^s \omega_t l(f(i_t, j_t), \mathbf{F})] \tag{7}$$

*where each $\omega_t$ takes values $\{\pm 1\}$ with equal probability. Then with the probability at least $1-p$, for all $f \in F_\Theta$ we have:*

$$\mathcal{R}_l(f) \leq \hat{\mathcal{R}}_l(f) + 2\mathbb{E}[\mathfrak{R}(F_\Theta)] + B\sqrt{\frac{\log\frac{1}{p}}{2s}}. \tag{8}$$

In order to upper-bound $\mathcal{R}_l$, both $\hat{\mathcal{R}}_l$ and model complexity $\mathbb{E}_\Omega[\mathfrak{R}(F_\Theta)]$ need to be upper-bounded. The next key lemma shows that what affect the model complexity term $\mathbb{E}_\Omega[\mathfrak{R}(F_\Theta)]$ in matrix completion context.

The Rademacher complexity can be bounded in terms of $\beta$ and $\gamma$ by the following lemma:

**Lemma 12** *Let $\mathcal{X} = \|\mathbf{X}\|_F$, $\mathcal{Y} = \|\mathbf{Y}\|_F$ and $d = \max(a,b)$,*

$$\mathbb{E}[\mathfrak{R}(F_\Theta)] \leq 2C_0 L_l\beta\mathcal{X}\mathcal{Y}\sqrt{\frac{\log 2d}{s}} + \sqrt{\frac{9dCL_l\alpha\sqrt{abp}(\sqrt{m}+\sqrt{n})}{s}}(s_1\mathcal{Y} + s_2\mathcal{X} + s_1 s_2) \tag{9}$$

For proving clearly we firstly introduce Lemma 13 as below, which is a special case of Theorem 2 in [3];

126 **Lemma 13** *Let $S_\sigma = \{\mathbf{W} \in \mathbb{R}^{n \times n} \mid \|\mathbf{W}\|_* \le \sigma\}$ and $a = \max_i \|\mathbf{A}_i\|_F$, where $\{\mathbf{A}_i \mid \mathbf{A}_i \in$*
127 $\mathbb{R}^{n \times n}\}_{i=1}^m$ *is an arbitrary set, then:*

$$\mathbb{E}[\sup_{W \in S_w} \frac{1}{m} \sum_{i=1}^m \omega_i \|\mathbf{W}\mathbf{A}_i\|_*] \le 2a\sigma \sqrt{\frac{\log 2n}{m}}. \tag{10}$$

128 By using Lemma 13 and Rademacher contraction principle(e.g. Lemma in [4]), we can readily prove
129 Lemma 12.

130 **Proof.** Denote $\mathbf{P} \in \mathbb{R}^{m \times n}$ with each entry $\mathbf{P}_{ij} = \sum_{\alpha:i_\alpha=i,j_\alpha=j} \omega_\alpha$, which means the 'hit-time'
131 on the $i,j$-th element of $\Omega$, then we can divide $\Re(F_\Theta)$ as:

$$\Re(F_\Theta) = \mathbb{E}_\sigma[\sup_{f \in F_\Theta} \frac{1}{s} \sum_{(i,j)} \mathbf{A}_{ij} l(f(i,j), \mathbf{F}_{ij})] + \mathbb{E}_\sigma[\sup_{f \in F_\Theta} \frac{1}{s} \sum_{(i,j)} \mathbf{B}_{ij} l(f(i,j), \mathbf{F}_{ij})] \tag{11}$$

In Eq. (11) we define

$$\mathbf{A}_{ij} = \begin{cases} \mathbf{P}_{ij}, & \text{if } h_{ij} > p \\ 0, & \text{otherwise.} \end{cases} \qquad \mathbf{B}_{ij} = \begin{cases} 0, & \text{if } h_{ij} > p \\ \mathbf{P}_{ij}, & \text{otherwise.} \end{cases}$$

132 where $h_{ij} = |\{\alpha : i_\alpha = i, j_\alpha = j\}|$ and $p$ is a thresholding value discussed soon. Recall that
133 $|l(f(i,j), \mathbf{F}_{ij})| \le B$, from Lemma 10 in [6] we can infer that:

$$\mathbb{E}_\sigma[\sup_{f \in F_\Theta} \frac{1}{s} \sum_{(i,j)} \mathbf{A}_{ij} l(f(i,j), \mathbf{F}_{ij})] \le \frac{B}{s} \mathbb{E}_\sigma[\sum_{(i,j)} |\mathbf{A}_{ij}|] \le \frac{B}{\sqrt{p}} \tag{12}$$

134 Also we need to bound the other term in Eq. 11 below by using Lemma 13. We conduct that

$$\mathbb{E}_\sigma[\sup_{f \in F_\Theta} \frac{1}{s} \sum_{(i,j)} \mathbf{B}_{ij} l(f(i,j), \mathbf{F}_{ij})]$$

$$\le \frac{L_l}{s} \mathbb{E}_\sigma[\sup_{\|\mathbf{G}\|_1 \le \alpha} \sum_{(i,j)} \mathbf{B}_{ij} \mathbf{x}_i^T \mathbf{G} \mathbf{y}_j + \sup_{\|\mathbf{G}\|_1 \le \alpha} \sum_{(i,j)} \mathbf{B}_{ij} \Delta\mathbf{x}_i^T \mathbf{G} \mathbf{y}_j + \tag{13}$$

$$\sup_{\|\mathbf{G}\|_1 \le \alpha} \sum_{(i,j)} \mathbf{B}_{ij} \mathbf{x}_i^T \mathbf{G} \Delta\mathbf{y}_j + \sup_{\|\mathbf{G}\|_1 \le \alpha} \sum_{(i,j)} \mathbf{B}_{ij} \Delta\mathbf{x}_i^T \mathbf{G} \Delta\mathbf{y}_j]$$

135 Since $\|\mathbf{G}\|_* \le C_0 \|\mathbf{G}\|_2 \le C_0 \|\mathbf{G}\|_1$ as the matrix-norm equivalence for any $\mathbf{G} \in \mathbb{R}^{a \times b}$ while there
136 always exists a fixed $C_0$, for the last three terms we can use Holder's inequality to upper-bound it as
137 below:

$$\frac{L_l}{s}[\sup_{\|\mathbf{G}\|_1 \le \alpha} \sum_{(i,j)} \mathbf{B}_{ij} \Delta\mathbf{x}_i^T \mathbf{G} \mathbf{y}_j + \sup_{\|\mathbf{G}\|_1 \le \alpha} \sum_{(i,j)} \mathbf{B}_{ij} \mathbf{x}_i^T \mathbf{G} \Delta\mathbf{y}_j + \sup_{\|\mathbf{G}\|_1 \le \alpha} \sum_{(i,j)} \mathbf{B}_{ij} \Delta\mathbf{x}_i^T \mathbf{G} \Delta\mathbf{y}_j]$$

$$\le \frac{L_l \mathbb{E}[\|\mathbf{B}\|_2]}{s} \sup_{\|\mathbf{G}\|_1 \le \alpha} [\|\Delta\mathbf{X}^T \mathbf{G} \mathbf{Y}\|_* + \sup_{\|\mathbf{G}\|_1 \le \alpha} \|\mathbf{X}^T \mathbf{G} \Delta\mathbf{Y}\|_* + \sup_{\|\mathbf{G}\|_1 \le \alpha} \|\Delta\mathbf{X}^T \mathbf{G} \Delta\mathbf{Y}\|_*]$$

$$\le \frac{\sqrt{ab}\alpha L \mathbb{E}[\|\mathbf{B}\|_2]}{s} [\|\Delta\mathbf{X}^T\|_F \|\mathbf{Y}\|_F + \|\mathbf{X}^T\|_F \|\Delta\mathbf{Y}\|_F + \|\Delta\mathbf{X}^T\|_F \|\Delta\mathbf{Y}\|_F] \tag{14}$$

$$\le \frac{\sqrt{ab}\alpha L}{s} (s_1 \mathcal{Y} + s_2 \mathcal{X} + s_1 s_2) \mathbb{E}[\|\mathbf{B}\|_2]$$

$$\le \frac{2.2 C L \alpha \sqrt{abp}(\sqrt{m} + \sqrt{n})}{s} (s_1 \mathcal{Y} + s_2 \mathcal{X} + s_1 s_2)$$

138 where the last inequality is from Lemma 1 in [6].

139 Next we bound the term $\mathbb{E}_\sigma[\sup_{\|\mathbf{G}\|_1 \le \alpha} \sum_{(i,j)} \mathbf{B}_{ij} \mathbf{x}_{i_\alpha}^T \mathbf{G} \mathbf{y}_{j_\alpha}]$ in Eq. (13) as:

$$
\begin{aligned}
\frac{L_l}{s} \mathbb{E}_\sigma[\sup_{\|\mathbf{G}\|_1 \le \alpha} \sum_{\alpha=1}^s \omega_\alpha \mathbf{x}_{i_\alpha}^T \mathbf{G} \mathbf{y}_{j_\alpha}] &\le L_l \mathbb{E}[\sup_{\|\mathbf{G}\|_1 \le \alpha} \frac{1}{s} \sum_{\alpha=1}^s \omega_\alpha tr(\mathbf{x}_{i_\alpha}^T \mathbf{G} \mathbf{y}_{j_\alpha})] \\
\le & L_l \mathbb{E}[\sup_{\|\mathbf{G}\|_1 \le \alpha} \frac{1}{s} \sum_{\alpha=1}^s \omega_\alpha tr(\mathbf{G} \mathbf{y}_{j_\alpha} \mathbf{x}_{i_\alpha}^T)] \le 2 C_0 L_l \alpha \max_{i,j} \|\mathbf{y}_j \mathbf{x}_i^T\|_2 \sqrt{\frac{\log 2d}{s}} \\
\le & 2 C_0 L_l \alpha \mathcal{X} \mathcal{Y} \sqrt{\frac{\log 2d}{s}}
\end{aligned}
\tag{15}
$$

140 Combining the above bounds in Eq. (12), Eq. (14) and Eq. (15) together, with $p =$
141 $(sB)/[2.2 CL\alpha\sqrt{abp}(\sqrt{m}+\sqrt{n})(s_1\mathcal{Y}+s_2\mathcal{X}+s_1s_2)]$ we can get the bound for $\mathbb{E}[\mathfrak{R}(F_\Theta)]$ as:
142

$$
\mathbb{E}[\mathfrak{R}(F_\Theta)] \le 2 C_0 L_l \alpha \mathcal{X} \mathcal{Y} \sqrt{\frac{\log 2d}{s}} + \frac{9 CL_l \alpha \sqrt{abp}(\sqrt{m}+\sqrt{n})}{s}(s_1\mathcal{Y}+s_2\mathcal{X}+s_1s_2)
\tag{16}
$$

143 ∎

144 Lemma 15 clarifies the upper-bound of the complexity of $f$. Additionally, with proper chosen $\lambda_E$
145 and $\lambda_G$, the empirical risk $\hat{\mathcal{R}}(f)$ can be sufficiently small. Therefore we conclude the upper bound
146 of $\mathcal{R}(f^*)$ as below.

**Lemma 14** *With a probability at least $1 - p$, thje expected l-risk of an optimal solution will be bounded by:*

$$
\mathcal{R}(f^*) \le 2 C_0 L_l \alpha \mathcal{X} \mathcal{Y} \sqrt{\frac{\log 2d}{s}} + \frac{18 CL_l \alpha \sqrt{abpN}}{s}(s_1\mathcal{Y}+s_2\mathcal{X}+s_1s_2) + B\sqrt{\frac{\log \frac{1}{p}}{2s}}
$$

147 Now consider another view to upper-bound our model, then we give Lemma 12 as followed,

148 **Lemma 15** *Let $\mathcal{X} = \|\mathbf{X}\|_F$, $\mathcal{Y} = \|\mathbf{Y}\|_F$ and $d = \max(a, b)$,*

$$
\mathbb{E}[\mathfrak{R}(F_\Theta)] \le 2 C_0 L_l [\gamma \sqrt{\frac{\log 2N}{s}} + \phi \sqrt{\frac{\log 2N}{s}} + \alpha \sqrt{\frac{\log 2d}{s}}(s_1\mathcal{Y}+s_2\mathcal{X}+s_1s_2)]
\tag{17}
$$

149 Again, by using Lemma 13 and Rademacher contraction principle(e.g. Lemma in [4]), we can prove
150 Lemma 15.

151 **Proof.** $\mathbb{E}(\mathfrak{R}(F_\Theta))$ can be bounded as above, we have

$$
\begin{aligned}
\mathbb{E}_\sigma[\sup_{f \in F_\Theta} \frac{1}{s} \sum_{\alpha=1}^s \omega_\alpha l(f(i_\alpha, j_\alpha), \mathbf{F}_{i_\alpha j_\alpha})] & \\
\le \frac{L_l}{s} \mathbb{E}_\sigma[\sup_{\|\mathbf{G}\|_1 \le \alpha} \sum_{\alpha=1}^s \omega_\alpha \mathbf{x}_{i_\alpha}^T \mathbf{G} \mathbf{y}_{j_\alpha} &+ \sup_{\|\mathbf{G}\|_1 \le \alpha} \sum_{\alpha=1}^s \omega_\alpha \Delta\mathbf{x}_{i_\alpha}^T \mathbf{G} \mathbf{y}_{j_\alpha} + \\
\sup_{\|\mathbf{G}\|_1 \le \alpha} \sum_{\alpha=1}^s \omega_\alpha \mathbf{x}_{i_\alpha}^T \mathbf{G} \Delta\mathbf{y}_{j_\alpha} &+ \sup_{\|\mathbf{G}\|_1 \le \alpha} \sum_{\alpha=1}^s \omega_\alpha \Delta\mathbf{x}_{i_\alpha}^T \mathbf{G} \Delta\mathbf{y}_{j_\alpha}]
\end{aligned}
\tag{18}
$$

Then one can follow the same approach in Eq. (15) as

$$\frac{L_l}{s}\mathbb{E}_\sigma[\sup_{\|\mathbf{G}\|_1\le\alpha}\sum_{\alpha=1}^s\omega_\alpha\mathbf{x}_i^T\mathbf{G}\mathbf{y}_j+\sup_{\|\mathbf{G}\|_1\le\alpha}\sum_{\alpha=1}^s\omega_\alpha\Delta\mathbf{x}_i^T\mathbf{G}\mathbf{y}_j+$$

$$\sup_{\|\mathbf{G}\|_1\le\alpha}\sum_{\alpha=1}^s\omega_\alpha\mathbf{x}_i^T\mathbf{G}\Delta\mathbf{y}_j+\sup_{\|\mathbf{G}\|_1\le\alpha}\sum_{\alpha=1}^s\omega_\alpha\Delta\mathbf{x}_i^T\mathbf{G}\Delta\mathbf{y}_j]$$

$$\le L_l\mathbb{E}[\frac{1}{s}(\sup_{\|\mathbf{E}\|_*\le\gamma}\sum_{\alpha=1}^s\omega_\alpha tr(\mathbf{E}_{i_\alpha j_\alpha}\mathbf{e}_{j_\alpha}\mathbf{e}_{i_\alpha}^T)+\sup_{\|\mathbf{\Phi}\|_F\le\phi}\sum_{\alpha=1}^s\omega_\alpha tr(\mathbf{\Phi}_{i_\alpha j_\alpha}\mathbf{e}_{j_\alpha}\mathbf{e}_{i_\alpha}^T))+ \tag{19}$$

$$2L_l\alpha r\sqrt{\frac{d\log 2d}{s}}[\max_{i,j}\|\mathbf{y}_j\Delta\mathbf{x}_i^T\|_2+\max_{i,j}\|\Delta\mathbf{y}_j\mathbf{x}_i^T\|_2+\max_{i,j}\|\Delta\mathbf{y}_j\Delta\mathbf{x}_i^T\|_2]$$

$$\le 2L_l[\gamma\sqrt{\frac{\log 2N}{s}}+C_0\phi\sqrt{\frac{\log 2N}{s}}+C_0\alpha\sqrt{\frac{\log 2d}{s}}(s_1\mathcal{Y}+s_2\mathcal{X}+s_1s_2)]$$

$$\le 2C_0L_l[\gamma\sqrt{\frac{\log 2N}{s}}+\phi\sqrt{\frac{\log 2N}{s}}+\alpha\sqrt{\frac{\log 2d}{s}}(s_1\mathcal{Y}+s_2\mathcal{X}+s_1s_2)]$$

where the last equation is derived by applying Lemma 13. So we derive another upper bound of $\mathbb{E}[\mathfrak{R}(F_\Theta)]$ as

$$\mathbb{E}[\mathfrak{R}(F_\Theta)]\le 2C_0L_l[\gamma\sqrt{\frac{\log 2N}{s}}+\phi\sqrt{\frac{\log 2N}{s}}+\alpha\sqrt{\frac{\log 2d}{s}}(s_1\mathcal{Y}+s_2\mathcal{X}+s_1s_2)] \tag{20}$$

∎

Then our Theorem 2 can be attained directly from Lemma 14 and Lemma 15.

**Theorem 2** *Denote $\|\mathbf{E}\|_*\le\alpha$, $\|\mathbf{G}\|_1\le\gamma$, and the perfect side feature matrices (containing latent features of $\mathbf{F}$) are corrupted with $\Delta\mathbf{X}$ and $\Delta\mathbf{Y}$ where $\|\Delta\mathbf{X}\|_F\le s_1, \|\Delta\mathbf{Y}\|_F\le s_2$ and $S = \max(s_1, s_2)$. To $\epsilon$-recover $\mathbf{F}$ that the expected loss $\mathbb{E}[l(f,\mathbf{F})]<\epsilon$ for a given arbitrarily small $\epsilon>0$, $O(\min((\gamma^2+\phi^2)\log N, S^2\alpha\sqrt{N})/\epsilon^2)$ observations are sufficient for our model to achieve an $\epsilon$-recovery when corrupted factors of side information are bounded.*

For the goal of investigating the recovery guarantee under the generalized frame of our work, it is noted that we can replace any norm-regularizers $\|\mathbf{G}\|_\sim$ of $\mathbf{G}$ satisfying that $\|\mathbf{G}\|_\sim\le\|\mathbf{G}\|_1$. Therefore it is feasible to further explore more structural priors in various situation.

# 3 Convergence Analysis

In this subsection, we present the proof of the global convergence for our algorithm.

For conveniently writing, we write the Lagrangian function of our problem as

$$\mathcal{L}(\mathbf{E},\mathbf{G},\mathbf{C},\mathbf{M}_1,\mathbf{M}_2,\beta)=\frac{1}{2}\|\mathbf{C}\|_F^2+\lambda_\mathbf{E}\|\mathbf{E}\|_*+\lambda_\mathbf{G}\|\mathbf{G}\|_1+$$

$$\langle\mathbf{M},\mathcal{B}(\mathbf{E})+\mathcal{A}(\mathbf{G})+\mathcal{N}(\mathbf{C})-\mathbf{D}\rangle+\frac{\beta}{2}\|\mathcal{B}(\mathbf{E})+\mathcal{A}(\mathbf{G})+\mathcal{N}(\mathbf{C})-\mathbf{D}\|_F^2 \tag{21}$$

where $\mathcal{B}(\mathbf{E})=\begin{pmatrix}\Omega(\mathbf{E})\\\mathbf{E}\end{pmatrix}$, $\mathcal{A}(\mathbf{G})=\begin{pmatrix}0\\-\mathbf{X}^T\mathbf{G}\mathbf{Y}\end{pmatrix}$, $\mathcal{N}(\mathbf{C})=\begin{pmatrix}0\\\mathbf{C}\end{pmatrix}$ and $\mathbf{D}=\begin{pmatrix}\Omega(\mathbf{F})\\0\end{pmatrix}$. $\mathbf{M}$ is the multiplier stacked as $\begin{pmatrix}\mathbf{M}_1\\\mathbf{M}_2\end{pmatrix}$.

The proving framework consists of three steps: The first step includes Lemma 16 for the proof of Lemma 17 and Theorem 3; the next step is the proof of Lemma 17 which indicates the convergence of our algorithm; the third step is to clarify our algorithm converges to a KKT point of problem (4), which is also the global minimizer for convex problem, shown in Theorem 3.

**Lemma 16** *Let $\mathbf{G}^k$, $\mathbf{E}^k$, $\mathbf{C}^k$ be the optimal solution for each individual subproblem at the $k$-th iteration, then it satisfies that $-\beta_k\tau_A(\mathbf{G}^{k+1}-\mathbf{G}^k)-\mathcal{A}^*(\bar{\mathbf{M}}^{k+1})\in\partial\|\mathbf{G}^{k+1}\|_1, -\beta_k\tau_B(\mathbf{E}^{k+1}-$*

176 $\mathbf{E}^k) - \mathcal{B}^*(\hat{\mathbf{M}}^{k+1}) \in \partial\|\mathbf{E}^{k+1}\|_*$ *where* $\bar{\mathbf{M}}^{k+1} = \mathbf{M}^k + \beta_k[\mathcal{A}(\mathbf{G}^k) + \mathcal{B}(\mathbf{E}^{k+1}) + \mathcal{N}(\mathbf{C}^k) - \mathbf{D}]$,

177 $\hat{\mathbf{M}}^{k+1} = \mathbf{M}^k + \beta_k[\mathcal{A}(\mathbf{G}^{k+1}) + \mathcal{B}(\mathbf{E}^{k+1}) + \mathcal{N}(\mathbf{C}^k) - \mathbf{D}]$, *here* $\partial\|\cdot\|$ *denotes the subgradient of*

178 *an arbitrary* $\|\cdot\|$, *and* $\mathcal{A}^*$ *is the adjoint operator of* $\mathcal{A}$.

179 Note that $\mathcal{A}^* = \mathbf{A}^T$ if $\mathcal{A}$ is a linear operator while $\mathcal{A}(\mathbf{X}) = \mathbf{A}\mathbf{X}$. This Lemma is directly derived
180 from the optimality conditions of subproblems when solving $\mathbf{G}$ and $\mathbf{E}$ individually.

181 Next we present the lemma implying the convergence.

182 **Lemma 17** *Given* $\beta_k$ *is non-decreasing and upper bounded,* $\tau_A > \|\mathcal{A}\|^2$, $\tau_B > \|\mathcal{B}\|^2$, *and*
183 $(\mathbf{G}^*, \mathbf{E}^*, \mathbf{C}^*, \mathbf{M}^*)$ *is any KKT point of problem 21, then:*

$$\{\tau_A\|\mathbf{G}^k - \mathbf{G}^*\|_F^2 - \|\mathcal{A}(\mathbf{G}^k - \mathbf{G}^*)\|_F^2 + \tau_B\|\mathbf{E}^k - \mathbf{E}^*\|_F^2 + \|\mathbf{C}^k - \mathbf{C}^*\|_F - \|\mathcal{N}(\mathbf{C}^k - \mathbf{C}^*)\|_F^2 +$$
$$\beta_k^{-2}\|\mathbf{M}^k - \mathbf{M}^*\|_F^2\} \text{ is non-increasing; and}$$
$$\|\mathbf{G}^k - \mathbf{G}^{k+1}\|_F^2 \to 0, \|\mathbf{E}^k - \mathbf{E}^{k+1}\|_F^2 \to 0, \|\mathbf{C}^k - \mathbf{C}^{k+1}\|_F^2 \to 0, \|\mathbf{M}^k - \mathbf{M}^{k+1}\|_F^2 \to 0. \quad (22)$$

184 For proving the non-increase property of the first sequence, it is equivalent to investigate the follow-
185 ing inequality:

$$\tau_A\|\mathbf{G}^{k+1} - \mathbf{G}^*\|_F^2 - \|\mathcal{A}(\mathbf{G}^{k+1} - \mathbf{G}^*)\|_F^2 + \tau_B\|\mathbf{E}^{k+1} - \mathbf{E}^*\|_F^2 + \|\mathbf{C}^{k+1} - \mathbf{C}^*\|_F$$
$$- \|\mathcal{N}(\mathbf{C}^{k+1} - \mathbf{C}^*)\|_F^2 + \beta_k^{-2}\|\mathbf{M}^{k+1} - \mathbf{M}^*\|_F^2 - (\tau_A\|\mathbf{G}^k - \mathbf{G}^*\|_F^2 - \|\mathcal{A}(\mathbf{G}^k - \mathbf{G}^*)\|_F^2 \quad (23)$$
$$+ \tau_B\|\mathbf{E}^k - \mathbf{E}^*\|_F^2 + \|\mathbf{C}^k - \mathbf{C}^*\|_F - \|\mathcal{N}(\mathbf{C}^k - \mathbf{C}^*)\|_F^2 + \beta_k^{-2}\|\mathbf{M}^k - \mathbf{M}^*\|_F^2) \le 0$$

186 For proving the above inequality, we list several facts to be used:

$$\mathbf{M}^{k+1} = \mathbf{M}^k + \beta_k(\mathcal{A}(\mathbf{G}^{k+1}) + \mathcal{B}(\mathbf{E}^{k+1}) + \mathcal{N}(\mathbf{C}^{k+1}) - \mathbf{D}),$$
$$2\left\langle \mathbf{G}^{k+1} - \mathbf{G}^*, \mathbf{G}^{k+1} - \mathbf{G}^k \right\rangle = \|\mathbf{G}^{k+1} - \mathbf{G}^*\|_F^2 - \|\mathbf{G}^k - \mathbf{G}^*\|_F^2 + \|\mathbf{G}^{k+1} - \mathbf{G}^k\|_F^2, \quad (24)$$
$$\mathcal{A}(\mathbf{G}^*) + \mathcal{B}(\mathbf{E}^*) + \mathcal{N}(\mathbf{C}^*) - \mathbf{D} = 0,$$
$$\langle \mathbf{M}, \mathcal{A}(\mathbf{G}) \rangle = \langle \mathcal{A}^*(\mathbf{M}), \mathbf{G} \rangle, \langle \mathbf{M}, \mathcal{B}(\mathbf{E}) \rangle = \langle \mathcal{B}^*(\mathbf{M}), \mathbf{E} \rangle.$$

**Proof.**

$$\tau_A\|\mathbf{G}^{k+1} - \mathbf{G}^*\|_F^2 - \|\mathcal{A}(\mathbf{G}^{k+1} - \mathbf{G}^*)\|_F^2 + \tau_B\|\mathbf{E}^{k+1} - \mathbf{E}^*\|_F^2 + \|\mathbf{C}^{k+1} - \mathbf{C}^*\|_F - \|\mathcal{N}(\mathbf{C}^{k+1} - \mathbf{C}^*)\|_F^2 +$$
$$\beta_k^{-2}\|\mathbf{M}^{k+1} - \mathbf{M}^*\|_F^2 - (\tau_A\|\mathbf{G}^k - \mathbf{G}^*\|_F^2 - \|\mathcal{A}(\mathbf{G}^k - \mathbf{G}^*)\|_F^2 + \tau_B\|\mathbf{E}^k - \mathbf{E}^*\|_F^2 + \|\mathbf{C}^k - \mathbf{C}^*\|_F$$
$$- \|\mathcal{N}(\mathbf{C}^k - \mathbf{C}^*)\|_F^2 + \beta_k^{-2}\|\mathbf{M}^k - \mathbf{M}^*\|_F^2)$$
$$=2\tau_A\left\langle \mathbf{G}^{k+1} - \mathbf{G}^*, \mathbf{G}^{k+1} - \mathbf{G}^k \right\rangle - \tau_A\|\mathbf{G}^{k+1} - \mathbf{G}^k\|_F^2 - 2\left\langle \mathcal{A}(\mathbf{G}^{k+1} - \mathbf{G}^*), \mathcal{A}(\mathbf{G}^{k+1} - \mathbf{G}^k) \right\rangle +$$
$$\|\mathcal{A}(\mathbf{G}^{k+1} - \mathbf{G}^k)\|_F^2 + 2\tau_B\left\langle \mathbf{E}^{k+1} - \mathbf{E}^*, \mathbf{E}^{k+1} - \mathbf{E}^k \right\rangle - \tau_B\|\mathbf{E}^{k+1} - \mathbf{E}^k\|_F^2 +$$
$$2\tau_N\left\langle \mathbf{C}^{k+1} - \mathbf{C}^*, \mathbf{C}^{k+1} - \mathbf{C}^k \right\rangle - \tau_A\|\mathbf{C}^{k+1} - \mathbf{C}^k\|_F^2 - 2\left\langle \mathcal{N}(\mathbf{C}^{k+1} - \mathbf{C}^*), \mathcal{N}(\mathbf{C}^{k+1} - \mathbf{C}^k) \right\rangle +$$
$$\|\mathcal{N}(\mathbf{C}^{k+1} - \mathbf{C}^k)\|_F^2$$
$$= - \{\beta_k^{-2}\|\mathbf{M}^{k+1} - \mathbf{M}^k\|_F^2 + \tau_B\|\mathbf{E}^{k+1} - \mathbf{E}^k\|_F - 2\beta_k^{-1}\left\langle \mathbf{M}^{k+1} - \mathbf{M}^k, \mathcal{B}(\mathbf{E}^{k+1} - \mathbf{E}^k) \right\rangle\} -$$
$$(\tau_A\|\mathbf{G}^{k+1} - \mathbf{G}^k\|_F^2 - \|\mathcal{A}(\mathbf{G}^{k+1} - \mathbf{G}^k)\|_F^2) - (\|\mathbf{C}^{k+1} - \mathbf{C}^k\|_F^2 - \|\mathcal{N}(\mathbf{C}^{k+1} - \mathbf{C}^k)\|_F^2) -$$
$$2\beta_k^{-1}\left\langle \mathbf{G}^{k+1} - \mathbf{G}^*, [-\beta_k\tau_A(\mathbf{G}^{k+1} - \mathbf{G}^k) - \mathcal{A}^*(\bar{\mathbf{M}}^{k+1})] + \mathcal{A}^*(\mathbf{M}^*) \right\rangle -$$
$$2\beta_k^{-1}\left\langle \mathbf{E}^{k+1} - \mathbf{E}^*, [-\beta_k\tau_B(\mathbf{E}^{k+1} - \mathbf{E}^k) - \mathcal{B}^*(\hat{\mathbf{M}}^{k+1})] + \mathcal{B}^*(\mathbf{M}^*) \right\rangle -$$
$$2\beta_k^{-1}\left\langle \mathbf{C}^{k+1} - \mathbf{C}^*, [-\beta_k(\mathbf{C}^{k+1} - \mathbf{C}^k) - \mathcal{N}^*(\mathbf{M}^{k+1})] + \mathcal{N}^*(\mathbf{M}^*) \right\rangle$$

$$(25)$$

Since $\tau_A \ge \|\mathcal{A}\|^2$, we can check that

$$\tau_A\|\cdot\|_F^2 - \|\mathcal{A}(\cdot)\|_F^2 \ge 0.$$

and similarly it is clear that

$$\beta_k^{-2}\|\mathbf{M}^{k+1} - \mathbf{M}^k\|_F^2 + \tau_B\|\mathbf{E}^{k+1} - \mathbf{E}^k\|_F^2 - 2\beta_k^{-1}\left\langle \mathbf{M}^{k+1} - \mathbf{M}^k, \mathcal{B}(\mathbf{E}^{k+1} - \mathbf{E}^k)\right\rangle \geq 0$$

The last three terms in Eq. (25) are nonnegative due to Lemma 16 and the monotonicity of subgradient mapping. So the non-increasing property in Lemma 17 is proved. Because of the non-increasing property and non-negativity, it has a limit. Then we can see that

$$\tau_A\|\mathbf{G}^{k+1} - \mathbf{G}^k\|_F^2 - \|\mathcal{A}(\mathbf{G}^{k+1} - \mathbf{G}^k)\|_F^2 \to 0,$$

$$\|\mathbf{C}^{k+1} - \mathbf{C}^k\|_F^2 - \|\mathcal{N}(\mathbf{C}^{k+1} - \mathbf{C}^k)\|_F^2 \to 0.$$

$$\beta_k^{-2}\|\mathbf{M}^{k+1} - \mathbf{M}^k\|_F^2 + \tau_B\|\mathbf{E}^{k+1} - \mathbf{E}^k\|_F^2 - 2\beta_k^{-1}\left\langle \mathbf{M}^{k+1} - \mathbf{M}^k, \mathcal{B}(\mathbf{E}^{k+1} - \mathbf{E}^k)\right\rangle \to 0$$

due to their non-negativity. So $\|\mathbf{G}^{k+1} - \mathbf{G}^k\|_F \to 0$ and $\|\mathbf{C}^{k+1} - \mathbf{C}^k\|_F \to 0$ can be obtained from the first two limits. Note that

$$\beta_k^{-2}\|\mathbf{M}^{k+1} - \mathbf{M}^k\|_F^2 + \tau_B\|\mathbf{E}^{k+1} - \mathbf{E}^k\|_F^2 - 2\beta_k^{-1}\left\langle \mathbf{M}^{k+1} - \mathbf{M}^k, \mathcal{B}(\mathbf{E}^{k+1} - \mathbf{E}^k)\right\rangle$$

$$\geq \beta_k^{-2}\|\mathbf{M}^{k+1} - \mathbf{M}^k\|_F^2 + \tau_B\|\mathbf{E}^{k+1} - \mathbf{E}^k\|_F^2 - 2\beta_k^{-1}\|\mathbf{M}^{k+1} - \mathbf{M}^k\|_F\|\mathcal{B}(\mathbf{E}^{k+1} - \mathbf{E}^k)\|_F$$

$$= (\beta_k^{-1}\|\mathbf{M}^{k+1} - \mathbf{M}^k\|_F - \|\mathcal{B}(\mathbf{E}^{k+1} - \mathbf{E}^k)\|_F)^2 + \tau_B\|\mathbf{E}^{k+1} - \mathbf{E}^k\|_F^2 - \|\mathcal{B}(\mathbf{E}^{k+1} - \mathbf{E}^k)\|_F^2$$

$$\geq \tau_B\|\mathbf{E}^{k+1} - \mathbf{E}^k\|_F^2 - \|\mathcal{B}(\mathbf{E}^{k+1} - \mathbf{E}^k)\|_F^2 \geq 0. \tag{26}$$

So we have that $\|\mathbf{E}^{k+1} - \mathbf{E}^k\|_F \to 0$. Furthermore,

$$\beta_k^{-2}\|\mathbf{M}^{k+1} - \mathbf{M}^k\|_F^2 + \tau_B\|\mathbf{E}^{k+1} - \mathbf{E}^k\|_F^2 - 2\beta_k^{-1}\left\langle \mathbf{M}^{k+1} - \mathbf{M}^k, \mathcal{B}(\mathbf{E}^{k+1} - \mathbf{E}^k)\right\rangle$$

$$(\beta_k^{-1}\|\mathbf{M}^{k+1} - \mathbf{M}^k\|_F - \sqrt{\tau_B}\|\mathbf{E}^{k+1} - \mathbf{E}^k\|_F)^2 +$$

$$2\beta_k^{-1}(\sqrt{\tau_B}\|\mathbf{M}^{k+1} - \mathbf{M}^k\|_F\|\mathbf{E}^{k+1} - \mathbf{E}^k\|_F - \left\langle \mathbf{M}^{k+1} - \mathbf{M}^k, \mathcal{B}(\mathbf{E}^{k+1} - \mathbf{E}^k)\right\rangle) \tag{27}$$

$$\geq (\beta_k^{-1}\|\mathbf{M}^{k+1} - \mathbf{M}^k\|_F - \sqrt{\tau_B}\|\mathbf{E}^{k+1} - \mathbf{E}^k\|_F)^2.$$

So $\beta_k^{-2}\|\mathbf{M}^{k+1} - \mathbf{M}^k\|_F^2 + \tau_B\|\mathbf{E}^{k+1} - \mathbf{E}^k\|_F^2 - 2\beta_k^{-1}\left\langle \mathbf{M}^{k+1} - \mathbf{M}^k, \mathcal{B}(\mathbf{E}^{k+1} - \mathbf{E}^k)\right\rangle \to 0$. This results in $\|\mathbf{M}^{k+1} - \mathbf{M}^k\|_F \to 0$ noting that $\|\mathbf{E}^{k+1} - \mathbf{E}^k\|_F \to 0$ . ∎

Based on Lemma 16 and Lemma 17, we can derive the following theorem.

**Theorem 3** *If $\beta_k$ is non-decreasing and upper-bounded, $\tau_A > \|\mathcal{A}\|$, and $\tau_B > \|\mathcal{B}\|$ then the sequence $\{(\mathbf{C}^k, \mathbf{G}^k, \mathbf{E}^k, \mathbf{M}^k)\}$ generated by adaptive LADMM converges to a KKT point of problem (4).*

**Proof.** By Lemma 17, $\{(\mathbf{C}^k, \mathbf{G}^k, \mathbf{E}^k, \mathbf{M}^k)\}$ is bounded, hence there is a subsequence that $(\mathbf{C}^{k_i}, \mathbf{G}^{k_i}, \mathbf{E}^{k_i}, \mathbf{M}^{k_i}) \to (\mathbf{C}^\infty, \mathbf{G}^\infty, \mathbf{E}^\infty, \mathbf{M}^\infty)$. We accomplish the proof in two steps.

We first prove that $(\mathbf{C}^\infty, \mathbf{G}^\infty, \mathbf{E}^\infty, \mathbf{M}^\infty)$ is a KKT point of our optimization problem.

By Lemma 17, $\mathcal{A}(\mathbf{G}^{k+1}) + \mathcal{B}(\mathbf{E}^{k+1}) + \mathcal{N}(\mathbf{C}^{k+1}) - \mathbf{D} = \beta_k^{-1}(\mathbf{M}^{k+1} - \mathbf{M}^k) \to 0$. This shows that any accumulation point of $\{(\mathbf{C}^k, \mathbf{G}^k, \mathbf{E}^k, \mathbf{M}^k)\}$ is a feasible solution.

Without the loss of generality, suppose $\lambda_G = \lambda_E = \frac{1}{2}$. by letting $k = k_i - 1$ in Lemma 16 and the subgradient definition, we have

$$\|\mathbf{G}^{k_i}\|_1 + \|\mathbf{E}^{k_i}\|_* + \|\mathbf{C}^{k_i}\|_F$$

$$\leq \|\mathbf{G}^*\|_1 + \|\mathbf{E}^*\|_* + \|\mathbf{C}^*\|_F + \left\langle \mathbf{G}^{k_i} - \mathbf{G}^*, -\beta_{k_i-1}\tau_A(\mathbf{G}^{k_i} - \mathbf{G}^{k_i-1}) - \mathcal{A}^*(\bar{\mathbf{M}}^{k_i})\right\rangle$$

$$+ \left\langle \mathbf{E}^{k_i} - \mathbf{E}^*, -\beta_{k_i-1}\tau_B(\mathbf{E}^{k_i} - \mathbf{E}^{k_i-1}) - \mathcal{B}^*(\hat{\mathbf{M}}^{k_i})\right\rangle + \left\langle \mathbf{C}^{k_i} - \mathbf{C}^*, -\beta_{k_i-1}(\mathbf{C}^{k_i} - \mathbf{C}^{k_i-1}) - \mathcal{N}^*(\mathbf{M}^{k_i})\right\rangle \tag{28}$$

Suppose $i \to \infty$, from Lemma 17, we can observe $\mathbf{G}^{k_i} - \mathbf{G}^{k_i-1} \to 0$ so that

$$
\begin{aligned}
&\|\mathbf{G}^\infty\|_1 + \|\mathbf{E}^\infty\|_* + \|\mathbf{C}^\infty\|_F^2 \\
\leq & \|\mathbf{G}^*\|_1 + \|\mathbf{E}^*\|_* + \|\mathbf{C}^*\|_F^2 + \langle \mathbf{G}^\infty - \mathbf{G}^*, -\mathcal{A}^*(\mathbf{M}^\infty)\rangle \\
& + \langle \mathbf{E}^\infty - \mathbf{E}^*, -\mathcal{B}^*(\mathbf{M}^\infty)\rangle + \langle \mathbf{C}^\infty - \mathbf{C}^*, -\mathcal{N}^*(\mathbf{M}^\infty)\rangle \\
= & \|\mathbf{G}^*\|_1 + \|\mathbf{E}^*\|_* + \|\mathbf{C}^*\|_F^2 - \langle \mathcal{A}(\mathbf{G}^\infty - \mathbf{G}^*), \mathbf{M}^\infty\rangle \\
& - \langle \mathcal{B}(\mathbf{E}^\infty - \mathbf{E}^*), \mathbf{M}^\infty\rangle - \langle \mathcal{N}(\mathbf{C}^\infty - \mathbf{C}^*), \mathbf{M}^\infty\rangle \\
= & \|\mathbf{G}^*\|_1 + \|\mathbf{E}^*\|_* + \|\mathbf{C}^*\|_F^2 - \langle \mathcal{A}(\mathbf{G}^\infty - \mathbf{G}^*) + \mathcal{B}(\mathbf{E}^\infty - \mathbf{E}^*) + \mathcal{N}(\mathbf{C}^\infty - \mathbf{C}^*), \mathbf{M}^\infty\rangle \\
= & \|\mathbf{G}^*\|_1 + \|\mathbf{E}^*\|_* + \|\mathbf{C}^*\|_F^2
\end{aligned}
\tag{29}
$$

since both $(\mathbf{C}^\infty, \mathbf{G}^\infty, \mathbf{E}^\infty)$ and $(\mathbf{C}^*, \mathbf{G}^*, \mathbf{E}^*)$ are feasible solutions. So we conclude that $(\mathbf{C}^\infty, \mathbf{G}^\infty, \mathbf{E}^\infty)$ is an optimal solution to (4).

Similarly we let $k = k_i - 1$ in Lemma 16 and by the definition of subgradient, we have

$$
\|\mathbf{G}\|_1 \geq \|\mathbf{G}^{k_i}\|_1 + \left\langle \mathbf{G} - \mathbf{G}^{k_i}, -\beta_{k_i-1}\tau_A(\mathbf{G}^{k_i} - \mathbf{G}^{k_i-1}) - \mathcal{A}^*(\bar{\mathbf{M}}^{k_i})\right\rangle
\tag{30}
$$

for any $\mathbf{G}$. Fix $\mathbf{G}$ and let $i \to \infty$, we see that

$$
\|\mathbf{G}\|_1 \geq \|\mathbf{G}^\infty\|_1 + \langle \mathbf{G} - \mathbf{G}^\infty, -\mathcal{A}^*(\mathbf{M}^\infty)\rangle
$$

for any $\mathbf{G}$. So $-\mathcal{A}^*(\mathbf{M}^\infty) \in \partial\|\mathbf{G}^\infty\|_1$. Similarly, $-\mathcal{B}^*(\mathbf{M}^\infty) \in \partial\|\mathbf{E}^\infty\|_*$. It is also not difficult to check that $-\mathcal{N}^*(\mathbf{M}^\infty) = \mathbf{C}$. Therefore, $(\mathbf{C}^\infty, \mathbf{G}^\infty, \mathbf{E}^\infty, \mathbf{M}^\infty)$ is a KKT point of problem (4).

Next we prove that the whole sequence of $\{(\mathbf{C}^k, \mathbf{E}^k, \mathbf{G}^k, \mathbf{M}^k)\}$ converges to $\{(\mathbf{C}^\infty, \mathbf{E}^\infty, \mathbf{G}^\infty, \mathbf{M}^\infty)\}$.

By choosing $(\mathbf{C}^*, \mathbf{G}^*, \mathbf{E}^*, \mathbf{M}^*) = (\mathbf{C}^\infty, \mathbf{G}^\infty, \mathbf{E}^\infty, \mathbf{M}^\infty)$ in Lemma 17, we have $\tau_A\|\mathbf{G}^{k_i} - \mathbf{G}^\infty\|_F^2 + \tau_B\|\mathbf{G}^k - \mathbf{G}^\infty\|_F^2 + \beta_{k_i}^{-2}\|\mathbf{M}^{k_i} - \mathbf{M}^\infty\|_F^2 \to 0$. By Lemma 17, we readily have $\tau_A\|\mathbf{G}^k - \mathbf{G}^\infty\|_F^2 - \|\mathcal{A}(\mathbf{G}^k - \mathbf{G}^\infty)\|_F^2 + \tau_B\|\mathbf{M}^k - \mathbf{M}^\infty\|_F^2 + \beta_k^{-2}\|\mathbf{M}^k - \mathbf{M}^\infty\|_F^2 \to 0$. So $(\mathbf{C}^k, \mathbf{G}^k, \mathbf{E}^k, \mathbf{M}^k) \to (\mathbf{C}^\infty, \mathbf{G}^\infty, \mathbf{E}^\infty, \mathbf{M}^\infty)$. Since $(\mathbf{C}^\infty, \mathbf{G}^\infty, \mathbf{E}^\infty, \mathbf{M}^\infty)$ can be an arbitrary accumulation point of $(\mathbf{C}^k, \mathbf{G}^k, \mathbf{E}^k, \mathbf{M}^k)$, we can conclude that $(\mathbf{C}^k, \mathbf{G}^k, \mathbf{E}^k, \mathbf{M}^k)$ converges to a KKT point. Since KKT point is the global optimal solution in the convex problem, $(\mathbf{C}^k, \mathbf{G}^k, \mathbf{E}^k, \mathbf{M}^k)$ converges to a global minimizer. ∎

# 4 Algorithm

In this section we establish the derivation for the closed-form solution of each subproblem. The four steps are noted as Updating $\mathbf{C}$, Updating $\mathbf{E}$, Updating $\mathbf{G}$ and Updating $\mathbf{M}$.

Updating $\mathbf{C}$:

$$
\mathbf{C}^{k+1} = \arg\min_{\mathbf{C}} \frac{1}{2}\|\mathbf{C}\|_F^2 + \left\langle \mathbf{M}_2^k, \mathbf{E}^k - \mathbf{X}^T\mathbf{G}^k\mathbf{Y} - \mathbf{C}\right\rangle + \frac{\beta_k}{2}\|\mathbf{E}^k - \mathbf{X}^T\mathbf{G}^k\mathbf{Y} - \mathbf{C}\|_F^2
\tag{31}
$$

which has a closed form solution as:

$$
\mathbf{C}^{k+1} = \frac{\beta_k}{\beta_k + 1}(\mathbf{E}^k - \mathbf{X}^T\mathbf{G}^k\mathbf{Y} + \mathbf{M}_2^k/\beta_k)
\tag{32}
$$

Updating $\mathbf{G}$:

$$
\min_{\mathbf{G}} \lambda_G\|\mathbf{G}\|_1 + \left\langle \mathbf{M}_2, \mathbf{E}^k - \mathbf{X}^T\mathbf{G}\mathbf{Y} - \mathbf{C}^k\right\rangle + \frac{\beta_k}{2}\|\mathbf{E}^k - \mathbf{X}^T\mathbf{G}\mathbf{Y} - \mathbf{C}^k\|_F^2,
\tag{33}
$$

after adding constant term to Eq. (33) we obtain

$$
\min_{\mathbf{G}} \lambda_G\|\mathbf{G}\|_1 + \frac{\beta_k}{2}\|\mathbf{B}^k - \mathbf{X}^T\mathbf{G}\mathbf{Y} - \mathbf{C}^k\|_F^2
\tag{34}
$$

where $\mathbf{B}_1^k = \mathbf{E}^k + \mathbf{M}_2^k/\beta_k$. By converting the matrix $\mathbf{b}$ into a vector $\mathbf{g} = \text{vec}(\mathbf{G})$, $\text{vec}(\mathbf{X}^T\mathbf{G}\mathbf{Y}) = (\mathbf{Y}^T \otimes \mathbf{X}^T)\mathbf{g}$. Further we let $\mathbf{b}^k = \text{vec}(\mathbf{B}^k)$ and $\otimes$ computes the Kronecker product of two matrices. Thus, if we denote $\mathbf{A} = (\mathbf{Y}^T \otimes \mathbf{X}^T)$, the above subproblem becomes:

$$
\min_{\mathbf{g}} \lambda_G\|\mathbf{g}\|_1 + \frac{\beta_k}{2}\|\mathbf{A}\mathbf{g} + \mathbf{c}^k - \mathbf{b}_1^k\|_2^2
\tag{35}
$$

Since (35) is a lasso problem, which does not have a closed-form solution and must be solved iteratively in practice, by utilizing a linearization technique, we have

$$\frac{1}{2}\|\mathbf{A}\mathbf{g} + \mathbf{c}^k - \mathbf{b}_1^k\|_2^2 \approx \frac{1}{2}\|\mathbf{A}\mathbf{g}^k + \mathbf{c}^k - \mathbf{b}_1^k\|_2^2 + \langle f_1^k, \mathbf{g} - \mathbf{g}^k \rangle + \frac{\tau_A}{2}\|\mathbf{g} - \mathbf{g}^k\|_2^2 \qquad (36)$$

where $\tau_A > 0$ is a proximal parameter and

$$f_1^k = \mathbf{A}^T(\mathbf{A}\mathbf{g}^k + \mathbf{c}^k - \mathbf{b}_1^k) = \mathbf{A}^T(\mathbf{A}\mathbf{g}^k + \mathbf{c}^k - \mathbf{e}^k - \mathbf{m}_2^k/\beta_k) \qquad (37)$$

is the gradient of $\frac{1}{2}\|\mathbf{A}\mathbf{g} + \mathbf{c}^k - \mathbf{b}_1^k\|_2^2$ at $\mathbf{g}_k$. Eq. (20) can be re-written as:

$$\min_{\mathbf{g}} \lambda_G \|\mathbf{g}\|_1 + \frac{\beta_k \tau_A}{2}\|\mathbf{g} - [\mathbf{g}^k - f_1^k/\tau_A]\|_2^2 \qquad (38)$$

Obviously the closed-form solution is:

$$\mathbf{g}^{k+1} = \max(|\mathbf{g}^k - f_1^k/\tau_A| - \frac{\lambda_G}{\tau_A \beta_k}, 0) \odot sgn(\mathbf{g}^k - f_1^k/\tau_A) \qquad (39)$$

Updating $\mathbf{E}$:

$$\begin{aligned} \min_{\mathbf{E}} \lambda_E \|\mathbf{E}\|_* + \Big\langle \mathbf{M}_1^k, R_\Omega(\mathbf{E} - \mathbf{F}) \Big\rangle + \frac{\beta_k}{2}\|R_\Omega(\mathbf{E} - \mathbf{F})\|_F^2 \\ + \Big\langle \mathbf{M}_2^k, \mathbf{E} - \mathbf{X}^T \mathbf{G}^{k+1}\mathbf{Y} - \mathbf{C}^k \Big\rangle + \frac{\beta_k}{2}\|\mathbf{E} - \mathbf{X}^T \mathbf{G}^{k+1}\mathbf{Y} - \mathbf{C}^k\|_F^2 \end{aligned} \qquad (40)$$

which we can reformulate as:

$$\min_{\mathbf{E}} \lambda_E \|\mathbf{E}\|_* + \frac{\beta_k}{2}\|R_\Omega(\mathbf{E} - \mathbf{B}_2^k)\|_F^2 + \frac{\beta_k}{2}\|\mathbf{E} - \mathbf{B}_3^k\|_F^2 \qquad (41)$$

where $\mathbf{B}_2^k = R_\Omega(\mathbf{F} - \mathbf{M}_1^k/\beta_k)$ and $\mathbf{B}_3^k = \mathbf{X}^T \mathbf{G}^{k+1}\mathbf{Y} + \mathbf{C}^k - \mathbf{M}_2^k/\beta_k$. After linearization, the problem can be approximately optimized by:

$$\min_{\mathbf{E}} \lambda_E \|\mathbf{E}\|_* + \frac{\beta_k \tau_B}{2}\|\mathbf{E} - (\mathbf{E}^k - f_2^k/\tau_B)\|_F^2 + \frac{\beta_k \tau_B}{2}\|\mathbf{E} - (\mathbf{E}^k - f_3^k/\tau_B)\|_F^2 \qquad (42)$$

where $f_2^k$ and $f_3^k$ are the gradients of $\frac{1}{2}\|R_\Omega(\mathbf{E} - \mathbf{B}_2^k)\|_F^2$ and $\frac{1}{2}\|\mathbf{E} - \mathbf{B}_3^k\|_F^2$ at $\mathbf{E}^k$, which are illustrated below:

$$\begin{aligned} f_2^k &= R_\Omega(\mathbf{E}^k - \mathbf{B}_2^k) = R_\Omega(\mathbf{E}^k - \mathbf{F} + \mathbf{M}_1^k/\beta_k), \\ f_3^k &= \mathbf{E}^k - \mathbf{B}_3^k = \mathbf{E}^k - \mathbf{X}^T \mathbf{G}^{k+1}\mathbf{Y} - \mathbf{C}^k + \mathbf{M}_2^k/\beta_k. \end{aligned} \qquad (43)$$

The closed-form solution is then readily obtainable as

$$\mathbf{E}^{k+1} = SVT(\mathbf{E}^k - (f_2^k + f_3^k)/(2\tau_B), \lambda_E/2(\beta_k \tau_B)) \qquad (44)$$

Here the operator $SVT(\mathbf{E}, t)$ is defined in [2] for soft-thresholding the singular values of an arbitrary matrix $\mathbf{E}$ by $t$.

Updating $\mathbf{M}$:

$$\begin{aligned} \mathbf{M}_1^{k+1} &= \mathbf{M}_1^k + \beta_k(R_\Omega(\mathbf{E}^{k+1} - \mathbf{F})), \\ \mathbf{M}_2^{k+1} &= \mathbf{M}_2^k + \beta_k(\mathbf{E}^{k+1} - \mathbf{X}^T \mathbf{G}^{k+1}\mathbf{Y} - \mathbf{C}^{k+1}). \end{aligned} \qquad (45)$$

# 5 Feature Description Table of Drug Discovery Dataset

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

Table 1: Drug corresponding feature.

| Label | Feature Name |
|-------|--------------|
| F1 | Molecular Weight( g/mol) |
| F2 | XLogP3 |
| F3 | Hydrogen Bond Donor Count |
| F4 | Hydrogen Bond Acceptor Count |
| F5 | Rotatable Bond Count |
| F6 | Exact Mass( g/mol) |
| F7 | Monoisotopic Mass( g/mol) |
| F8 | Topological Polar Surface Area |
| F9 | Heavy Atom Count |
| F10 | Complexity |
| F11 | Defined Atom Stereocenter Count |
| F12 | Undefined Atom Stereocenter Count |
| F13 | Defined Bond Stereocenter Count |
| F14 | Covalently-Bonded Unit Count |
| F15 | Universal |

[3] S. M. Kakade, K. Sridharan, and A. Tewari. On the complexity of linear prediction: Risk bounds, margin bounds, and regularization. In *Advances in neural information processing systems*, pages 793–800, 2009.

[4] R. Meir and T. Zhang. Generalization error bounds for bayesian mixture algorithms. *The Journal of Machine Learning Research*, 4:839–860, 2003.

[5] B. Recht. A simpler approach to matrix completion. *The Journal of Machine Learning Research*, 12:3413–3430, 2011.

[6] O. Shamir and S. Shalev-Shwartz. Matrix completion with the trace norm: Learning, bounding, and transducing. *Journal of Machine Learning Research*, 15:3401–3423, 2014.

[7] M. Xu, R. Jin, and Z. hua Zhou. Speedup matrix completion with side information: Application to multi-label learning. *Advances in Neural Information Processing Systems 26*, pages 2301–2309, 2013.