[Reviews · NeurIPS 2016]

Reviewer 1

Summary

This paper proposes a new inductive matrix completion method that extends the common interactive model X’GY to approximate the target F by introducing an intermediate low rank E matrix that is the estimation of F, while adding L1 norm regularizer over G. Theoretical and empirical results are presented. It is trivial to extend the common model to include linear terms, which is not significant enough to be claimed as one major contribution. The approach is new mostly because it adds the sparsity regularization over G instead of the low-rank constraint. However empirically, it is not clear how effective is the sparsity regularizer. The theoretical results seem to be obtained in the very ideal case. The empirical study is not convincing.

Qualitative Assessment

1.Some important notations are not introduced in the paper. It does not introduce the r, N, \mathcal{X} used in Theorem 1 and 2. These notations are critical and they should be given in the paper instead of asking readers to find them in some other places. 2.The theoretical results in this paper show a \epsilon-recovery sample complexity of O(logN). However this is based the assumption that r=O(1) (for Theorem 1) and \alpha = O(1) (for Theorem 1 and 2). These assumptions are very strong. In general, it could be in the order of r, \alpha = O(N). Especially, on real world data, the small constant assumption is too ideal. 3. Experiments: (1) The RMSE measure used in the paper is problematic. According to the equation given in the paper, it measures the relative squared error on observed entries. But one really should measure the error on unobserved entries. (2) Only two real-world datasets are used in the experiments. Both matrices are very small (943 x 1682 and 46 x 26). More datasets in larger sizes should be considered. (3) It is not clear how were the trade-off parameters are determined in the experiments for all the comparison methods. For example, for the proposed method, how to choose the values for \lambda_G and \lambda_E? (4) For the results reported, are them averages of multiple runs? If so, how many runs and what are the standard deviations?

Confidence in this Review

2-Confident (read it all; understood it all reasonably well)


Reviewer 2

Summary

This paper focuses on the problem of matrix completion, with additional side information. The paper formulates the problem as a convex optimization problem, and gives a theoretical results on the sample complexity for achieving an exact recovery. Besides, the paper also gives some stability analysis when the side matrix is contaminated. The paper proposes an algorithm based on linearized alternating direction method (LADMM) and show that it works well in simulations and real data sets.

Qualitative Assessment

My main main concern is about the novelty—compared with previous works, it seems that the main difference is a slightly different objective function (essentially this work proposes a different regularization). That being said, this work might still be considered important if the applications are important, since it is shown to outperform previous methods in real-world data sets. Some more questions and comments: 1. the paper does not mention the probability space and how Ω is chosen in Theorem 1. I guess the probability space is from “observed entries are uniformly sampled at random”. 2. The theoretical analysis requires X and Y to be full row rank (or approximately full row rank), which essentially exclude the cases that X and Y are well-conditioned “tall and thin” matrix, i.e., when m is larger than a and n is larger than b. I wonder if anything can be said to this case, to if there is any reason that this case is not considered. 3. It would be helpful for to add some discussions about Theorem 1 on its strength, for example, what are the expected magnitude of μ and q0. 4. It is unclear how to choose function g() and the regularization parameters λ in the simulations. == post-rebuttal update== About the novelty, other reviewer pointed out that there is some novelty in the model of low-rank part. In stead of assuming that a low-rank matrix, it is assumed to a product of three matrices: XGY, where X and Y are known and G is a sparse matrix. Considering it, I revised "Novelty/originality" from 2 to 3. However, it is still unclear to me that what is the motivation in assuming the sparsity of G.

Confidence in this Review

2-Confident (read it all; understood it all reasonably well)


Reviewer 3

Summary

The authors present a new model for matrix completion with side information. The novelty of the approach is that the core parameter of the model, the matrix G, is not assumed to be low-rank but to be sparse. The associated algorithm is proposed and experimental results show the advantage of the method it synthetic as well as real datasets.

Qualitative Assessment

My only concern is the evaluation with synthetic data. The authors state (P6L254) that the synthetically generated matrix G is full rank. And then the method that encourage G to be low rank do not recover the proper G. Well, of course. The comparison in this case is completely unfair. Please, re-do the experiments with synthetic data (also?) with low-rank G's. Otherwise the methods will never be able to recover G. The authors provided 17 pages of supplementary material containing 17 lemmas and 3 theorems. Well, if all of this is needed to proof the main result of the paper, perhaps NIPS is not the best venue given the space limitations. More recent computer-vision applications of matrix completion can be found in: - Alameda-Pineda, X., Yan, Y., Ricci, E., Lanz, O., & Sebe, N. (2015, October). Analyzing free-standing conversational groups: a multimodal approach. In Proceedings of the 23rd ACM international conference on Multimedia (pp. 5-14). ACM. - Alameda-Pineda, X., Ricci, E., Yan, Y., & Sebe, N. (2016). Recognizing Emotions From Abstract Paintings Using Non-Linear Matrix Completion. In Proceedings of the IEEE Conference on Computer Vision and Pattern Recognition (pp. 5240-5248). Minor comments - P1L24 M_ij = 1 - P1L19: classically, - P2L54: determinEs - P3L111: what is ||G||_2? Is it different from the Frobenius norm? - P3L138: the second line of the equation is wrong, since the matrix M does not appear in the rhs - P4L145: with high probability (without a) - P5L198: we propose an adaptive (check your spelling) - P5L200: I do not understand what do you mean by guarantee the performance. - P5L204: group of variables (without the) and subproblem (without s) - P6L233: It looks that t_a and t_b appear because beta_k is non-decreasing and upper-bounded, but this two bounds exist because A and B are linear operators. - P6L242-245: It is not clear if the exact same observation is given to all the methods. - P6L254: You can say N(0,100) instead of multiplying by 100. - P7L271: low ranK - P8L313: I do not understand this sentence. If the problem is convex the algorithm can converge? Well of course, many algorithms converge if the problem is convex. I do not understand the interest of this sentence. - Unify references 19 and 20. - Where was 22 published. - What is the publication year of 24?

Confidence in this Review

3-Expert (read the paper in detail, know the area, quite certain of my opinion)


Reviewer 4

Summary

The authors study matrix completion with side information (e.g., user information and item information). Let X, Y and F be a user feature matrix, an item feature matrix and a rating matrix to complete, respectively. The authors consider a bi-linear model of the form F \approx X^T G Y, where G is a parameter matrix. Unlike existing approaches, which impose a low-rank constraint on G, this paper imposes a sparse constraint on G and a low-rank constraint on X^T G Y. The paper presents exact/approximate recovery guarantees and a linearized ADMM algorithm for parameter estimation.

Qualitative Assessment

Major comments -------------- * Perhaps a more explicit title would be "A Sparse Interactive Model for Matrix Completion with Side Information" * Theorem 3: it was not clear why a new analysis is needed compared to [28] or [19]. Are there novel contributions or is it included for completeness? * Experimental results show that the proposed method outperform existing methods by a large margin. The difference (especially on Movielens) is so large that it makes me question whether hyper-parameter tuning was really carried out seriously. In particular, how was the rank determined for IMC and DirtyIMC? Since low-rank G is not preferred, have the authors tried setting high rank? * Have you tried to compare with just a Frobenius norm constraint on G? Since the number of users and items is not too large in the experiments, G should fit in memory. * The solvers used for the existing methods are not clear. For IMC, did you use non-convex or convex optimization? A short summary of IMC, DirtyIMC and MAXIDE would be welcome too. * In practice, this approach could run into troubles if the number of user and item features are both very high and G is not sufficiently sparse. Detailed comments ----------------- * Line 13: > An efficient linearized Lagrangian algorithm is developed with a strong guarantee of convergence. This is too vague. Please make a more precise statement. * Line 27: You may also want to cite: A. K. Menon and C. Elkan. Link prediction via matrix factorization. In Machine Learning and Knowledge Discovery in Databases, pages 437–452. 2011. * Line 92: "We propose" is misleading, since this has been done before. * Line 140: Please also remind what r is. * Line 198: "we propose" -> "we employ" or "we propose to use"? * Equation (35): Instead of "(35) does not have a closed-form solution due to a linear operator A", you could mention that (35) is a lasso problem and must be solved iteratively in practice. * Line 241: > For all methods, hyperparameters were tuned 241 via the same cross validation process. which is? * Figures: use vector graphics if possible * Line 271: > These results sugested that incorporating the strong prior of low ran G might hurt the prediction/recovery performance If the ground truth G is full-rank, this is not really surprising. Please clearly specify the rank of G on line 251. Typos ----- * Line 54: determins -> determines * Line 125: impaire -> impair

Confidence in this Review

2-Confident (read it all; understood it all reasonably well)


Reviewer 5

Summary

This paper studies the problem of low-rank matrix completion with side information: Given P_{\Omega}(E_0) (i.e., a part of E_0), recover E_0, where it is assumed that E_0 = X^T G_0 Y (where X and Y encode some side information). Unlike previous studies, which often assumes that G_0 to be low-rank, this work is exploring the sparsity of G_0. Due to this key idea, the proposed method can archive a sampling compelxity as low as O(r logN), the lowest complexity I have seen. In addition, the authors also devise an ADM based optimization algorithm to solve the raised minimization problem (2). Convergence analysis is provided too.

Qualitative Assessment

Novelty: It is novel to consider the sparsity (other than low-rankness) of G. The theocratical results are novel too. Potential impact: This work has some solid contributions to the community of low-rank matrix completion/recovery. But the idea of side information has been studied by many others. Clarity: It seems that the paper is prepared in rush. The writing of this paper (particularly the Appendix) has many minor issues, e.g., 1) Line 134, it is better to define E_0 = U \Sigma V^T instead of F = U \Sigma V^T. BTW, it is abnormal to assume E_0 = F. Anyway, it is unnecessary to use two symbols to denote the same thing. 2) Line 137~139, for any M, we cannot have P_{T} (M) + P_{T^\bot}(M) = M. So, it is better to write instead "for any M that satisfies M = P_X M P_Y". 3) Appendix, Line 19, the second P_{T} should be P_{T^\bot} 4) Appendix, Line 20, Condition A2 --- > Condition A1. 5) Appendix, Line 23, \|E + \Delta E\|_* + \lambda \|G_0 + \Delta G\|_1 = \|G_0 + \Delta G\|_* + \lambda \|G_0 + \Delta G\|_1 is incorrect. But this could be fixed by deleting " = \|G_0 + \Delta G\|_* + \lambda \|G_0 + \Delta G\|_1 " 6) Appendix, above and below Line 24, \|UV^T + U_{\bot}V_{\bot}^T\|_F = 1 is incorrect. But this could be fixed by changing \|UV^T + U_{\bot}V_{\bot}^T\|_F to \|UV^T + U_{\bot}V_{\bot}^T\| (operator norm). 7) Appendix, Line 31, "Since ... which is implied from assumption A2" --> this sentence should be placed at the beginning of Line 28. 8) \lambda s >= \lambda \|G_0\|_1 --> \lambda s = \lambda \|G_0\|_1 9) Appendix, the second equation below Line 38 should be ">=", not "=" 10) Appendix, Line 44, \Delta G_{t} should be G_0 + \Delta G_{t} 11) Appendix, Lemma 2, why it is required that |\Omega| \leq T_1 ? 12) Appendix, Line 55, r < < a --> r < = a. 13) Appendix, the equation right above Line 70: P_T R_{\Omega} P_T (M) should be < M, P_T R_{\Omega} P_T (M) > I have not checked the contents since Line 73. The authors are required to proof-read the paper, in particular the Appendix section. 14) There are some recent papers closely related to this work and should be included for discussions: a) Low-rank matrix completion in the presence of high coherence, IEEE T-SP, 2016. b) Robust Principal Component Analysis with Side Information, ICML 2016. 15) In the possible journal version of this paper, the reader want to see the extension to the case where the few observed entries are contaminated by noise.

Confidence in this Review

2-Confident (read it all; understood it all reasonably well)


Reviewer 6

Summary

This paper addresses the problem of matrix completion when the side information about the features is available in form of features vector matrices X and Y. In such problems the matrix to be completed can be written as F = X^T G Y. The paper considers a model in which the matrix F is low rank and G is sparse. They propose a recovery method that minimizes the nuclear norm of X^T G Y and \ell_1 norm of G. This seems to be a fairly intuitive approach to complete the matrices that follow this model. The paper also proposes a solution along the lines of regularized maximum likelihood formulation for the matrix completion under Gaussian noise assumptions. Finally, the authors propose a LADMM algorithm for solving the problem and provide a proof of its convergence. Overall, the paper looks fairly exhaustive in the theoretical analysis of the proposed solution.

Qualitative Assessment

The main comments are as follows: 1) The problem formulation under noise given in equation (2) of the paper seems to be problematic. It is not clear where the noise enters in the overall setup. Generally, the observations are corrupted by noise, i.e., R_{\Omega}(F) = R_{\Omega} (X^T G Y + N). Under such a scenario, the likelihood assuming gaussian noise is given by || R_{\Omega}(F) - R_{\Omega} (X^T G Y ) ||_F^2. The paper however does something confusing. It introduces a matrix E, and introduces a constraint R_{\Omega}(E) = R_{\Omega}(F) and uses the likelihood term ||E - X^T G Y||_F^2. This is problematic because the estimated low rank matrix E is constrained to have same value as the noisy observation. This basically will eliminate the possibility of de-noising the observed part of the matrix. As per my knowledge of existing literature for noisy matrix completion problem, the observed entries of the estimated matrix are never constrained to be equal to their noisy observations. 2) The main recovery result given in Theorem 2 introduces model assumptions ||E||_* <= \alpha and ||G||_1 <= \gamma. These assumptions raise some questions whose answers are not well argued in the paper. Firstly, even though bounds on any norm of the model parameters makes it inherently less complex, bounded nuclear norm and \ell_1 norm does not necessarily imply low rank E and sparse G. The whole paper is built upon the notion that G is sparse and E is low rank but the assumption in theorem does not necessary impose these on the model. In this regard looking at the Theorem statement it is not clear why authors minimize weighed sum nuclear norm and \ell_1 norm. Under the model assumptions stated in the Theorem a straight forward alternative is to consider constrained maximum likelihood formulation, i.e., put the constraints ||E||_* <= \alpha, ||G||_1 <= \gamma explicitly in the recovery problem formulation. A discussion based on these points will significantly improve the clarity of the paper from motivational point of view. 3) It is not clear how do the recovery results in the Theorem statement depend on the regularization parameters \lambda_E and \lambda_G. As the value of regularization parameters increases the error should increase for very large values of these parameters. But this is not evident from the recovery bounds. 4) In the final LADMM formulation it is not clear why authors introduce an auxiliary variable C. Even without C, the G update step and E update are similar. For clarity authors need to explain why C is needed. Minor Comments: 1) In abstract, it is not clear what is the size of the matrix. Mentioning that it is N = max{n_1, n_2} where n_1 is the number of rows and n_2 is number of columns. 2) Typo ’impaire’ on line 125. 3) In statement of Theorem 2, \Chi^2 is not defined. 4) After Theorem 3, a comment on how to calculate \tau_a and \tau_b would help the reader to use the proposed algorithm.

Confidence in this Review

2-Confident (read it all; understood it all reasonably well)